# Laplacian Spectra of Persistent Structures in Taiwan, Singapore, and US Stock Markets

**DOI:** 10.3390/e25060846

**Published:** 2023-05-25

**Authors:** Peter Tsung-Wen Yen, Kelin Xia, Siew Ann Cheong

**Affiliations:** 1Center for Crystal Researches, National Sun Yat-sen University, 70 Lienhai Rd., Kaohsiung 80424, Taiwan; 2School of Physical and Mathematical Sciences, Nanyang Technological University, 21 Nanyang Link, Singapore 637371, Singapore

**Keywords:** graph laplacian, stock market, complex systems, persistent structure, Fiedler vector

## Abstract

An important challenge in the study of complex systems is to identify appropriate effective variables at different times. In this paper, we explain why structures that are persistent with respect to changes in length and time scales are proper effective variables, and illustrate how persistent structures can be identified from the spectra and Fiedler vector of the graph Laplacian at different stages of the topological data analysis (TDA) filtration process for twelve toy models. We then investigated four market crashes, three of which were related to the COVID-19 pandemic. In all four crashes, a persistent gap opens up in the Laplacian spectra when we go from a normal phase to a crash phase. In the crash phase, the persistent structure associated with the gap remains distinguishable up to a characteristic length scale $\epsilon^{*}$ where the first non-zero Laplacian eigenvalue changes most rapidly. Before $\epsilon^{*}$, the distribution of components in the Fiedler vector is predominantly bi-modal, and this distribution becomes uni-modal after $\epsilon^{*}.$ Our findings hint at the possibility of understanding market crashs in terms of both continuous and discontinuous changes. Beyond the graph Laplacian, we can also employ Hodge Laplacians of higher order for future research.

## 1. Introduction

Unlike simple systems, where we can easily identify the few relevant variables and deduce the mathematical equations that they must obey (conservation laws, equations of state, equations of motion), or for thermodynamic systems, where we identify extensive and intensive variables that are statistical sums and averages of the microscopic variables, for complex systems it is difficult to identify a set of simplified (coarse-grained) variables [1,2]. This is especially challenging, since we know that self-organization and emergence is a hallmark of complex systems, implying that the effective variables might change from time to time [3]. One of the directions explored by complex systems scientists is to embed the N variables onto a low-dimensional manifold, using information contained in their time series $X_{i=1,\ldots ,N}\left(t\right)$ [4,5]. Recently, D’Addese et al. [6] and Villani et al. [7] used information-theoretic methods to identify the *relevant sets of variables* in random Boolean networks, gene-regulatory networks, MAPK signaling pathways in eukaryotes, and other systems, and the manifold they evolve on. Others have turned instead to topological data analysis (TDA) and persistent homology to achieve the same goal [8,9]. Still others have combined information-theoretic methods and simplicial complexes arising from TDA to identify effective variables, and their interactions in the form of higher-order networks [10].

To be useful for describing a complex system, effective variables must change slowly with time, so that we do not need to switch between different sets of effective variables frequently. Of the N≫1 microscopic variables, we find some combinations that change dramatically over short time scales, as well as other combinations that evolve slowly. We call the former *fast variables*, and the latter *slow variables* [11,12]. Frequently, the slow variables do not evolve independently, but form groups that co-evolve. These are then persistent structures that are consistent with self-organization (in that their equations of motion are not built into the microscopic dynamics) and emergence (the groups themselves can vary over long times) in the complex system. The first step towards understanding how we should write down the effective variables would be to identify the persistent structures. We attempted to do this in our two previous papers on TDA and Ricci curvature analysis (RCA). In our first paper [8], we applied TDA to identify persistent structures in financial correlation networks during market crashes. This attempt is an extension of our exploration into financial market dynamics using more traditional econophysics methods such as the minimal spanning tree (MST) [13,14,15,16,17,18,19], and the planar maximally filtered graph (PMFG) [20,21,22,23,24]. We were attracted to TDA because it can give us more information than graph filtering methods, as illustrated by how the Betti numbers change in toy models where two shells merge through the formation of a bottleneck, or when a shell changes into a torus through intermediate spindle torus and horn torus stages. However, the computation of persistent Betti numbers is tedious and time-consuming, and generally not feasible at large length scales. At smaller length scales, the number of persistent structures is large, making it impossible to identify all of them automatically.

More importantly, in TDA two persistent structures are assumed to have become one, the moment they become connected by a neck. As illustrated in Figure 1, we believe that persistent structures remain distinguishable beyond this first connection, so long as we can tell them apart from the neck region connecting them. Therefore, in our second paper [9], we introduced tools from Ricci curvature to help identify persistent structures with positive Ricci curvatures nearly everywhere, and neck regions with negative Ricci curvatures. By following the evolution of a particular neck over a market crash, we visualized how it was formed (down to the exact component stocks) and destroyed. Nevertheless, challenges remain. First, RCA is not easy to implement and automate. Second, small curvature changes are hard to detect because they involve collective movements of many nodes. To this end, new perspectives and approaches are necessary for the elucidation of the overall dynamical picture.

Drawing upon our experience in studying undergraduate physics, we can solve problems more easily by changing our approach or rephrasing our questions from a different perspective. In solid state physics, we find concepts such as the Brillouin zone, band structure, Fermi level, and band gap emerging naturally when we choose to work in momentum space. Additionally, owing to the band theory of solids so obtained we can predict such emergent phases as conductors, semi-conductors, half-metals, and insulators. In our two TDA papers, we investigated financial correlations in real space by examining simplicial complexes obtained through the filtration process. Here, we make a first attempt at characterizing such correlations in “momentum space”. Before we dive into the spectral analysis of financial correlations, we first explain what persistent structures are and how to think of their continuous and discontinuous changes in Section 2, by using a raindrop analogy. Thereafter, in Section 3, we briefly review the filtration procedure in TDA, before arguing for the theoretical connection between symmetries and block-diagonal matrices. In particular, in solid state physics, the symmetries are in real space while the block-diagonal matrices appear in momentum space, whereas for networks or simplicial complexes, diagonal blocks associated with community structure appears in real space, and thus we expect the symmetries to be in momentum space. Communities in networks or simplicial complexes are normally discovered from adjacency matrices $A_{ij}$, but they can also be discovered from the graph Laplacians $L_{ij}$, which has interpretations closer to the Hamiltonian matrix $H_{ij}$ in quantum mechanics, and their spectral properties are better understood. In the remainder of Section 3, we illustrate using various toy models of community structures that the existence of persistent clusters separated in space show up as a persistent gap in the spectra of $L_{ij}$. From the Fiedler eigenvector, associated with the first non-zero eigenvalue $\lambda_{1}$ of $L_{ij}$, we can identify the neck, in addition to the persistent clusters. We also realize from these studies that the persistent clusters remain distinct even after they become linked, up till the point where $\lambda_{1}$ changes most rapidly with change in length scale. In Section 4, we apply these insights to analyze the correlations in real-world stock markets, by sliding six-month time windows across four market crashes on three stock exchanges, to see how the topology and geometry of such correlations change with time. We found the existence of two distinct phases in stock markets. In the normal phase, the spectrum of Laplacian eigenvalues has no gaps (consistent with the market being a single giant cluster), whereas in the crash phase, we find a gap emerging at large length scales (consistent with the market breaking into two or more clusters). Finally, we conclude in Section 5.

## 2. Intuition on Persistent Structures

Before we formally define persistent structures in Section 3, let us first develop an intuition on these based on a familiar physical phenomenon. In an atmosphere saturated with water vapor, water droplets can nucleate around impurities. When a water droplet first forms, it is small and light, and can be suspended by warm air rising from the earth’s surface. The water droplet can then lose mass through evaporation, or it can absorb more water vapor from the atmosphere to become larger and heavier. Eventually, it becomes too heavy to be suspended by the rising warm air and begins to fall toward the earth’s surface as a raindrop. As the raindrop falls, it rubs against the air and deforms into the characteristic teardrop shape (Figure 2a). Even though the raindrop now consists of a large number of water molecules (Figure 2b), it continues to lose water molecules through evaporation (Figure 2c), or gain water molecules through absorption (Figure 2d). More importantly, as the raindrop gains speed falling through air, its surface becomes unstable. The trailing end of the raindrop may then breakup into smaller droplets (Figure 2e).

Instead of microscopic water molecules, we prefer to describe the phenomenon in terms of raindrops. This is because many raindrops retain their identities as they descend to the earth’s surface. Indeed, if we perform instantaneous hierarchical clustering on the collection of water molecules coming down as rain, each raindrop is a robust cluster at a convenient length scale. However, unlike robust clusters with constant compositions, the compositions of raindrops change across length scale and time. It is thus better to think of a raindrop as a persistent homological structure, from the TDA point of view. Persistent homological structures need not have fixed compositions with respect to changes in length scale and time. They just need to have the same set of defining topological characteristics. For example, when a “sphere” comprising 20 particles grows over time to become one having 1000 particles, we can continue to refer to the structure as a “sphere” ($\beta_{0}=1$), provided it has no holes ($\beta_{1}=0$) and no voids ($\beta_{2}=0$).

Indeed, in this analogy, the raindrop 10 km above ground has a composition different from the raindrop that reaches the ground. Nevertheless, we think of the two as the same raindrop at different times, because it can be tracked continuously from an altitude of 10 km down to the ground. On the other hand, if an old raindrop completely evaporates at a height $h_{1}$, and thereafter a new raindrop suddenly forms at height $h_{2}<h_{1}$, we do not consider the new raindrop to be the same persistent structure as the old raindrop. Therefore, a change in composition is admissible for a persistent structure, provided this change is always slow.

Treating the raindrop as a persistent structure and ignoring its compositional changes, we can then describe the time evolution of the raindrop in terms of the position $\vec{R}\left(t\right)$ of its centre of mass, and its volume V(t). The former is the mean
(1)$$\vec{R}\left(t\right)=\frac{1}{N}\overset{N}{\underset{i=1}{\sum }}{\vec{r}}_{i}\left(t\right)$$
of the N≫1 water molecules making up the raindrop, while the latter is related to the covariances
(2)$$\left[\begin{array}{ccc} \Sigma_{xx} & \Sigma_{xy} & \Sigma_{xz}\\ \Sigma_{yx} & \Sigma_{yy} & \Sigma_{yz}\\ \Sigma_{zx} & \Sigma_{zy} & \Sigma_{zz} \end{array}\right]=[\begin{array}{ccc} \frac{1}{N}\overset{N}{\underset{i=1}{\sum }}{\left(x_{i}-X\right)}^{2} & \frac{1}{N}\overset{N}{\underset{i=1}{\sum }}\left(x_{i}-X\right)\left(y_{i}-Y\right) & \frac{1}{N}\overset{N}{\underset{i=1}{\sum }}\left(x_{i}-X\right)\left(z_{i}-Z\right)\\ \frac{1}{N}\overset{N}{\underset{i=1}{\sum }}\left(y_{i}-Y\right)\left(x_{i}-X\right) & \frac{1}{N}\overset{N}{\underset{i=1}{\sum }}{\left(y_{i}-Y\right)}^{2} & \frac{1}{N}\overset{N}{\underset{i=1}{\sum }}\left(y_{i}-Y\right)\left(z_{i}-Z\right)\\ \frac{1}{N}\overset{N}{\underset{i=1}{\sum }}\left(z_{i}-Z\right)\left(x_{i}-X\right) & \frac{1}{N}\overset{N}{\underset{i=1}{\sum }}\left(z_{i}-Z\right)\left(y_{i}-Y\right) & \frac{1}{N}\overset{N}{\underset{i=1}{\sum }}{\left(z_{i}-Z\right)}^{2} \end{array}],$$
where $\left(X,Y,Z\right)=\vec{R}$. Of course, the shape of the raindrop can also change with time. This is determined by the higher-order statistical moments of ${\left\{\left(x_{i}\left(t\right),y_{i}\left(t\right),z_{i}\left(t\right)\right)\right\}}_{i=1}^{N}$. However, we can only adopt this hierarchical description in terms of position, size, shape, … provided the topological characteristics of the raindrop remains unchanged. If the raindrop breaks up into two raindrops, or if an air bubble forms within the raindrop, our description of the first raindrop would have to change discontinuously.

Through this analogy, we hope to convince our readers that persistent structures are the most convenient variables to develop physical theories around. A persistent structure is a collection of microscopic variables that is long-lived (temporal persistence), insensitive to changes in length scales (spatial persistence), and whose statistical moments change continuously with time. The last requirement is guaranteed by topological persistence, i.e., the Betti numbers $\beta_{0}$, $\beta_{1}$, … remaining constant.

## 3. Formal Spectral Definition of Persistent Structures

### 3.1. TDA Definition of Persistence

In Section 2, we saw that a raindrop remains well-defined as a persistent structure over the time it takes to fall to the ground. Therefore, within this time, we can write down equations that govern the continuous changes in its position, velocity, size, and shape. This description is useful because the raindrops are well separated in space. In contrast, the description of a swimming pool in terms of water droplets is not useful, first because there is no natural size to use for such water “droplets”, and second because slight “movements” of these “droplets” would make them overlap with each other (and lose their distinctive identities). A discontinuous change occurs when two “droplets” merge, and therefore the structures before and after merging cannot be treated as the same. The structure before does not persist past the merger, while the structure after does not exist until the merger.

It is this spatial persistence that the filtration procedure in TDA identifies. As shown in Figure 3, we draw a link between two data points at filtration parameter ϵ, if their pairwise distance is less than or equal to ϵ. We then write the network obtained in terms of a simplicial complex, which is a set consisting of 0-simplices (nodes), along with 1-simplices (links), 2-simplices (triangles), along with higher-order k-simplices, which are complete graphs with (*k* + 1) nodes. We can also define a *face* of a k-simplex to be a (k−1)-simplex making up the *k*-simplex, and the set of all faces of a k-simplex its *boundary*. In terms of these constructs, a simplicial complex Σ can be precisely defined as a set of simplices satisfying two conditions: (1) any face of a simplex in Σ is also in Σ; and (2) the intersection of any two simplices $\sigma_{1}$ and $\sigma_{2}$ in Σ is either the empty set ∅, or a face of both $\sigma_{1}$ and $\sigma_{2}.$ As ϵ is increased, we find more connected components in Σ. For example, at $t_{1}$ and $\epsilon_{1}$ in Figure 3, the simplicial complex obtained is $\Sigma_{1}=\left\{\langle 1,9\rangle ,\langle 5,8\rangle ,\langle 1\rangle ,\ldots ,\langle 9\rangle \right\}$, which has only two 1-simplices (〈1,9〉 and 〈5,8〉) and no 2-simplices, whereas at $t_{1}$ and $\epsilon_{2}>\epsilon_{1}$, the simplicial complex $\Sigma_{2}=\left\{\langle 1,7,4\rangle ,\langle 1,6,7\rangle ,\ldots ,\langle 2,10,5\rangle ,\langle 1,4\rangle ,\langle 1,6\rangle ,\ldots ,\langle 5,8\rangle ,\langle 1\rangle ,\ldots ,\langle 9\rangle \right\}$ obtained has six 2-simplices (〈1,7,4〉,〈1,6,7〉,…,〈2,10,5〉) and 14 1-simplices. At $t_{1}$ and $\epsilon_{3}$, the simplicial complex obtained is $\Sigma_{3}=\left\{\langle 1,3,6,7\rangle ,\langle 2,3,5,10\rangle ,\langle 2,5,8,10\rangle ,\langle 1,7,4\rangle \ldots ,\langle 5,10,8\rangle ,\langle 1,4\rangle ,\ldots ,\langle 5,8\rangle ,\langle 1\rangle ,\ldots ,\langle 9\rangle \right\}$. In this example, 〈i〉 is a 0-simplex, 〈i,j〉 a 1-simplex, 〈i,j,k〉 a 2-simplex, and 〈i,j,k,l〉 a 3-simplex.

To follow the dynamics, we start at the smallest scale $\epsilon_{1}$, to find 6 isolated nodes (0-simplices) and 2 links each connecting 2 nodes (1-simplices) at $t_{1}$. At this same scale, we have 7 isolated 0-simplices, 1 1-simplex consisting of 2 links connecting 3 nodes at $t_{2}$, as well as 3 0-simplices and 3 1-simplices (2 of them consisting of 1 link connecting 2 nodes, and 1 of them consisting of 2 links connecting 3 nodes) and $t_{3}$. In contrast, at the intermediate scale $\epsilon_{2}$, we find a connected simplicial complex with 10 0-simplices, 14 1-simplices, and 6 2-simplices at time $t_{1}$. At the scale $\epsilon_{2}$, and time $t_{2}$, the simplicial complex has two connected components. The first consists of 5 0-simplices and 4 1-simplices. The second consists of 5 0-simplices, 7 1-simplices, and 3 2-simplices. Finally, at $t_{3}$, the connected simplicial complex at scale $\epsilon_{2}$ has 10 0-simplices, 14 1-simplices, and 5 2-simplices.

Not all connected components identified through the filtration process are persistent, because they remain topologically distinct over very small ranges of ϵ. When TDA was first invented, it was applied onto data sets obtained at one point in time or averaged over time. Therefore, the range $\left(\epsilon_{b},\;\epsilon_{d}\right)$ between the scale $\epsilon_{b}$ a topologically distinct component first appears (also called the *birth* of the component) and the scale $\epsilon_{d}$ it disappears (also called the *death* of the component) is referred to as its *lifetime*. In TDA, the lifetimes of components are typically shown in the form of a *barcode* or a *persistence* diagram. In a barcode (see Figure 4a), each bar shows the birth (component first appears) and the death (component disappears) of a component in the simplicial complex as ϵ is varied. In a persistence diagram (see Figure 4b), a component is represented as a point whose x coordinate is the birth time, and whose y coordinate is the death time. Persistent components must have long lifetimes, and we can identify these by looking in the barcode in Figure 4a for bars that are significantly longer than the previous ones (the last two bars), or large deviations from the diagonal in the persistence diagram. In the example shown in Figure 4b, there are two 0-dimensional components with lifetimes greater than ϵ=1.0. These two merged into one at $\epsilon_{d}=1.58$, compared to the most recent death at ϵ≲0.5, and can therefore be thought of as persistent components. In contrast, none of the 1-dimensional components shown in Figure 4b are persistent.

In the example shown in Figure 4, the data set contains two persistent clusters by construction. When there are more persistent structures at different length scales, identifying them from barcodes and persistence diagrams will become challenging. In the rest of this section, we will show that it is easier, and more systematic, to identify persistent structures in spectral space. In fact, this was first demonstrated by Donath and Hoffmann [25], as well as Fiedler [26], who identified communities based on the eigenvectors of the adjacency matrix and the Laplacian matrix respectively. We refer readers to the survey Spielman and Teng [27], and the tutorial on spectral clustering by von Luxburg [28]. To understand why spectral clustering works so well, let us start with what we know about block-diagonal matrices in quantum mechanics.

### 3.2. Block-Diagonal Matrices in Quantum Mechanics

The barcodes and persistence diagrams described in Section 3.1 are visualizations in real space. It turns out that we can also identify persistent structures in spectral space. To do this, we start from the adjacency matrix representation of the simplicial complex. As we show in Section 3.3.1, there are no persistent structures for a single cluster of data points. Thus, the simplest example that can help us understand how persistent structures are identified would be two well-separated clusters of data points in Section 3.3.2. The adjacency matrix thus has a well-defined community structure, with one diagonal block for the first cluster, and a second diagonal block for the second cluster.

In quantum mechanics, we were first introduced to block-diagonal matrices when we explore the implications of symmetries. For example, we know that the angular momentum operator $L^{2}$ and $L_{z}$ (the *z*-component of the angular momentum) have the same eigenvectors |l m〉, with eigenvalues $L^{2}\left|l\;m\rangle =l\left(l+1\right)\hbar^{2}\right|l\;m\rangle$ and $L_{z}\left|l\;m\rangle =m\hbar \right|l\;m\rangle .$ Since m=−l,−l+1,…,0,…,l−1,l, the matrix representation of $L^{2}$ is organized into (2l+1)×(2l+1) diagonal blocks (see Figure 5a). We were taught that this is the consequence of a symmetry, embodied by the commutation relation $\left[L^{2},L_{z}\right]=0,$ with the diagonal blocks being irreducible representations of this symmetry. In this angular momentum example, the diagonal blocks do not have the same sizes. In contrast, in solid state physics, the diagonal blocks have the same sizes. To see this, consider a crystal made up of $N=N_{1}N_{2}N_{3}$ repeating unit cells. At the boundary of this crystal, we apply the Born-von Karman boundary conditions, to write the wave function as $\psi \left(\vec{r}\right)=\psi \left(\vec{r}+N_{1}{\vec{a}}_{1}+N_{2}{\vec{a}}_{2}+N_{3}{\vec{a}}_{3}\right)$, where ${\vec{a}}_{1}$, ${\vec{a}}_{2}$, ${\vec{a}}_{3}$ are the primitive lattice vectors. Furthermore, the periodic crystal has translational symmetry, and thus $\psi \left(\vec{r}+\vec{R}\right)=e^{i\vec{k}\cdot \vec{R}}\psi \left(\vec{r}\right).$ Therefore, when we Fourier transform the Hamiltonian matrix in real space, we obtain a Hamiltonian matrix in momentum space that is block-diagonal (see Figure 5b). Each N×N diagonal block is associated with a distinct wave vector $\vec{k}$. Diagonalizing the block for $\vec{k}$, we would obtain widely separated energy eigenvalues $E_{1}\left(\vec{k}\right),\;E_{2}\left(\vec{k}\right),\;\ldots ,E_{n}\left(\vec{k}\right),\ldots$. Similarly, from the diagonal blocks of ${\vec{k}}^{\prime }$ and ${\vec{k}}^{\prime \prime }$, we obtain the energy eigenvalues $\left\{E_{n}\left({\vec{k}}^{\prime }\right)\right\}$ and $\left\{E_{n}\left({\vec{k}}^{\prime \prime }\right)\right\}$. As shown in Figure 5c, $E_{n}\left(\vec{k}\right)$, $E_{n}\left({\vec{k}}^{\prime }\right)$, and $E_{n}\left({\vec{k}}^{\prime \prime }\right)$ have comparable values, and thus when we combine $E_{n}\left(\vec{k}\right)$ for all values of $\vec{k}$, we obtain the nth energy band of the crystal. Between the nth energy band and the (n+1)th energy band of the crystal, we find the nth band gap for the band structure of the crystal.

For a network with adjacency matrix A, we have $A_{ij}=1$ if node i is linked to node j, or $A_{ij}=0$ otherwise. In general, nodes in the network need not have the same degree k, i.e., $k_{i}\neq k_{j}$ for nodes i≠j. These node degrees can be computed from A, as $k_{i}=\overset{N}{\underset{j=1}{\sum }}A_{ij}$, and thereafter organized into a degree matrix $K=\mathrm{diag}\left(k_{1},\ldots ,k_{N}\right)$. In terms of A and K, the *graph Laplacian* can be defined as L=K−A. In Figure 5d, we show the adjacency matrix A (or equivalently the Laplacian matrix L) of a network with well-defined communities (no overlaps between communities). For such a network, A or L would also be block diagonal. The diagonal block associated with community c would be N(c)×N(c), where N(c) is the number of nodes in community c. Treating the Laplacian L as the Hamiltonian of the network, this block-diagonal structure tells us that there is an observable C that commutes with L, i.e., [L, C]=0, and thus the community structure represents some sort of symmetry. More importantly, given the block-diagonal structure of L, its eigenvalues would also be organized into bands separated by band gaps. One of the first to observe these bands of Laplacian eigenvalues separated by a gap was Arenas [29].

### 3.3. Analysis of Spectral Sequence, Overlapping Communities, Persistent Structures

In the filtration process of a given data set, we vary ϵ to obtain networks with different link densities. When ϵ is small, we expect isolated data points and small clusters of data points. The network is largely unconnected, and therefore we obtain a distribution of eigenvalues for small clusters. As ϵ increases, larger clusters start to form, looking initially like star networks, but eventually becoming complete networks. From spectral graph theory, which is the study of the properties of a network in terms of its characteristic polynomial, eigenvalues $\left\{\lambda_{i}\right\}$, and eigenvectors $\left\{{\vec{u}}_{i}\right\}$ of L [30,31], we know that any connected component will have one eigenvector with λ=0. If a network of N nodes consists of M connected components, then each of the components would contribute one zero eigenvector, i.e., λ=0 would be M-fold degenerate. Over and above the zero eigenvalue, special networks such as a star network with N nodes has N−2 unit eigenvalues λ=1, and one eigenvalue λ=N, whereas a complete network with N nodes has instead N−1 eigenvalues λ=N. For real networks with intermediate link densities, we then expect the unit eigenvalues λ=1 to shift progressively to λ=N as the link density increases. This tells us that as link density increases, the distribution of eigenvalues becomes more concentrated at larger eigenvalues. This is illustrated in Figure 6.

When ϵ=40 in Figure 6a, only the two data points closest to each other are linked, whereas the rest of the data points remain isolated. Here, we find the Laplacian eigenvalue λ=0 being seven-fold degenerate, and the eigenvalue λ=2 for the cluster with two nodes. When the filtration parameter is increased to ϵ=65 in Figure 6b, the eight data points become a fully connected network. However, the connectivity is not uniform across the network, and part of it looks like a less-densely linked star, while the other more-densely linked part consists of connected 2-simplices. For this oddly shaped network, and also the one shown in Figure 6c when ϵ=90, the nonzero eigenvalues are distributed between $\lambda_{\min}$ and $\lambda_{\max}.$ Finally, when the filtration parameter reaches λ=115 in Figure 6d, three nodes attain the maximum degree of $k_{\max}=7$. This is why the maximum eigenvalue $\lambda_{\max}=8$ is three-fold degenerate. We call the spectral space visualization $\left\{\lambda_{n}\left(\epsilon \right)\right\}$ shown in Figure 6 as ϵ is varied in the filtration process a *spectral sequence*. In the following subsections, we show the spectral sequences of different simple configurations of data points, to identify the relevant features characterizing these configurations. We also show how these features change with separation between clusters, during a fusion process, and in the presence of noise of different strengths.

#### 3.3.1. One Cluster

As a benchmark, let us examine the spectral sequence of a single cluster of data points sampled from a two-dimensional Gaussian distribution. As we can see from Figure 7, there is no prominent band gap in the spectral sequence. We use spectral graph theory to explain this in two limits. First, in the limit of small ϵ, the simplicial complex consists of multiple connected components of different sizes $n_{i}$. Furthermore, if these connected components are networks intermediate between star and complete networks, their eigenvalues would be distributed between λ=0 and $\lambda =n_{i}.$ When we superimpose these spectra, we find a “continuous” distribution of eigenvalues between λ=0 and $\lambda =\underset{i}{\max}n_{i}.$ Second, in the limit of large ϵ, the simplicial complex consists of a single connected component intermediate between star and complete networks of size N. Therefore, the nonzero Laplacian eigenvalues of such a simplicial complex would also be “continuously” distributed between $\lambda_{\min}>1$ and $\lambda_{\max}<N$.

#### 3.3.2. Two Clusters

The simplest example of a data set with community structure would be one with two clusters, as shown in Figure 8. The barcode of this data set was shown in Figure 4, where we saw that this two-cluster structure is persistent with respect to changes in length scale. From Figure 8, we see that there are two zero eigenvalues from ϵ≈0.7 to ϵ≈1.8. This range of filtration parameter is comparable to the one found from the barcode in Figure 4. However, the spectral signature ($\Delta \lambda =\underset{i}{\max}\left\{\lambda_{i+1}-\lambda_{i}\right\}$, shaded yellow in Figure 8) for this persistent structure is far more prominent, suggesting that the two clusters remain distinguishable even after links start to form between them (overlapping communities). In particular, when ϵ=2.8653 and $\lambda_{1}=12.9498$, there are 261 links between the two clusters, but Δλ remains larger than level spacings elsewhere in the spectrum.

Starting at ϵ=1.9254, the two clusters become linked, and there is only one zero eigenvalue $\lambda_{0}=0$. At this filtration parameter, the first nonzero eigenvalue is $\lambda_{1}=0.6055$. For a smooth manifold M, Jeff Cheeger first proved that $\lambda_{1}\geq h^{2}\left(M\right)/4$, where $\lambda_{1}$ is the first nonzero eigenvalue of the Laplace–Beltrami differential operator on M, while the Cheeger constant h(M) is the smallest area of a hypersurface that cuts M into two [32]. This result carries over to discrete networks. Suppose $\lambda_{1}$ is the first nonzero eigenvalue of the Laplacian of network G, which can be split into two networks A (with $N_{A}$ nodes) and B (with $N_{B}$ nodes) by cutting the smallest number of links $n_{AB}$, then $\lambda_{1}\geq h^{2}\left(G\right)/4$, where $h\left(G\right)=n_{AB}/\max\left(N_{A},N_{B}\right)$ [33,34,35]. This tells us that $\lambda_{1}$ increases with the size of the neck linking networks A and B.

### 3.4. Analysis of Eigenvectors, and Identification of the Neck from the Fiedler Vector

The *Fiedler vector* ${\vec{u}}_{1}$ associated with $\lambda_{1}$ also allows us to identify nodes that are part of the neck [26,36]. In this subsection, we show how this can be done, by first showing the results from toy networks before we analyze the Fiedler vector and other low-lying eigenvectors in the spectral sequence examples shown in Section 3.3 and Appendix A.

#### 3.4.1. Toy Networks

In Table 1, we show a sequence of toy networks in which two distinguishable subnetworks are connected by necks of various natures. In the first two networks, the clusters share an edge or a corner, and thus the neck consists of the nodes making up the edge or the corner. In the next two networks, the clusters are bridged by a single node or an edge, and thus the neck consists of the bridging node(s). Nodes in the neck can be identified as zero components in the Fiedler vector. We can also distinguish the two clusters, because the components in one of them is positive, while the other is negative. In the last network, the two clusters are not balanced, with one consisting of four nodes, and the other three nodes. In its Fiedler vector, the weight of the neck (node 5) is not zero, but still significantly smaller than the weights of the other nodes. Through these examples, we realized that the neck consists of nodes with weights close to zero, or significantly smaller than the clustered nodes in the Fiedler vector.

#### 3.4.2. Filtration Sequence for Two Clusters

In Section 3.3 and Appendix A, we analyzed spectral sequences resulting from the filtration of different data sets, to identify tell-tale signatures for different numbers of clusters. For the spectral sequence shown in Figure 3 for two clusters of data points, let us focus on the Fiedler vectors for ϵ=1.9254 ($\lambda_{1}=0.6055$) and ϵ=3.3353 ($\lambda_{1}=29.704$). At ϵ=1.9254, the two clusters are connected by 9 links, between 5 nodes from cluster 1, and 5 nodes from cluster 2. These 10 nodes, identified from their *smaller absolute weights*, form the neck between clusters 1 and 2. For ϵ=3.3353, there are 21 nodes with zero weights. All 21 nodes have the maximum degree $k_{i}=49$ in a network of 50 nodes and are members of a *bloated neck*.

Just to be careful, we also look at the node in cluster 2 with the minimum degree $k_{i}=33$. This node has the largest absolute weight in ${\vec{u}}_{1}$, and is linked to all cluster-2 nodes, but only to 14 cluster-1 nodes. Out of these 14 cluster-1 nodes, 13 of them belong to the neck. In addition, we find that set of 10 neck nodes when ϵ=1.9254 is a subset of the set of 21 neck nodes when ϵ=3.3353. This tells us that in the filtration process, instead of a simple fusion A+B→C, TDA suggests the process A+B→A+n+B→a+N+b→N=C. In other words, the fusion between clusters A and B begin with the creation of a small neck n. This neck continues to absorb members of A and B to become the bigger neck N (at the expenses of clusters A→a and B→b shrinking), until all original members of clusters A and B become absorbed into N, which we can now call cluster C.

In this example, we examine the filtration process involving two clusters. However, we expect the picture to hold even for the filtration processes at different times for two clusters merging into one, since the neck should be present until the two clusters completely fuse together. However, the smaller neck at an earlier time may not be embedded within the larger neck later. This is because even necks can lose or gain nodes, and all processes described in the raindrop analogy apply.

#### 3.4.3. Quasi-Degeneracies and Multiple Necks

Finally, we consider the situation where the data points are connected by more than one neck at some stage in the filtration process. The simplest situation where this occurs is when we have three clusters along a straight line, as shown in Figure 9a. In Figure 9b, we see that when ϵ=0.668, the three clusters are not linked, and we find three zero eigenvalues. When the filtration parameter is increased to ϵ=1.328, the three clusters forms a single cluster, with one neck connecting the green cluster to the blue, and another neck connecting the blue cluster to the red. When this occurs, $\lambda_{1}=1.067$ and $\lambda_{2}=2.776$ become non-zero, but remain close to each other. In Figure 9c,d, we show that the eigenvectors associated with $\lambda_{1}$ and $\lambda_{2}$ are the antisymmetric and symmetric combinations of the green and red clusters respectively.

In the symmetric combination ${\vec{u}}_{2}$, components of the green and red clusters have the same sign, thus forcing components of the middle blue cluster to have the opposite sign. Components of these three clusters can only have the same sign in ${\vec{u}}_{0}$, the eigenvector associated with $\lambda_{0}=0$. In the antisymmetric combination ${\vec{u}}_{1}$, components of the green and red clusters have opposite signs, and thus components of the middle blue cluster must be close to zero. In this sense, although a cluster in its own right, in the antisymmetric combination ${\vec{u}}_{1}$ the blue cluster plays the role of a neck. Because of this dual role, we call the blue cluster a *bridging cluster*. Because of these differences in signs and magnitudes, if we divide the component $u_{1,i}$ by $u_{2,i}$, this ratio would be close to zero if i is a member of the blue cluster (the bridging cluster, 0≤i<40), or has an absolute value close to one if i is a member of the green cluster (40≤i<70) or the red cluster (70≤i<100). Indeed, this is what we see in Figure 9e. In Figure 9e, we also see four absolute ratios that are exceptionally large. This can only happen if $u_{2,i}$ is close to zero, but $u_{1,i}$ is not. From Figure 9d, we see that these four are true members of the two necks connecting the three clusters. Indeed, when we go to ϵ=1.987 in Figure 9f, where $\lambda_{1}=10.645$ and $\lambda_{2}=23.791$ are very different, $r_{i}\approx 0$ continues to help us identify the bridging cluster, while $\left|r_{i}\right|\gg 1$ helps us identify the neck, which is thicker at this filtration parameter.

### 3.5. Spectral Definition of Persistent Structure

Summarizing our findings from Section 3.3 and Section 3.4 and Appendix A, we realized that persistent structures are accompanied by persistent gaps $\Delta \lambda =\underset{i}{\max}\left(\lambda_{i+1}-\lambda_{i}\right)$ in the spectral sequence. These persistent gaps should not be confused with non-persistent ones that appear when the persistent structures have a discrete spectrum of sizes. From Figure 8, we see that this persistent gap arises because $\lambda_{2}$ increases more rapidly than $\lambda_{1}$ (which can remain zero) when ϵ was first increased, before $\lambda_{1}$ increases rapidly after ϵ exceeded the characteristic gap between the most persistent clusters. In particular, when $\lambda_{1}$ starts rising, the persistent structures are already connected by necks, but they remain distinguishable, i.e., we can talk about A+n+B (a thin neck n connecting two large clusters A and B) or a+N+b (a thick neck N connecting two small clusters a and b). We think of the persistent structures A and B as having vanished only after they are completely absorbed by the neck N, at which time we can identify it as a new persistent structure C where all nodes from *A* and *B* have become a complete network. This picture is confirmed by our analysis of the Fiedler eigenvector ${\vec{u}}_{1}$ (corresponding to $\lambda_{1}>0$). From the eigenvector perspective, the persistent structures remain distinguishable from the neck since nodes in the neck have zero or smaller absolute weights in the Fiedler vector compared to nodes in the clusters.

Through Section 3.3 and Section 3.4 and Appendix A, we now have a deeper appreciation of the raindrop analogy described in Section 2. Clearly, when two persistent structures A and B are not connected, their individual descriptions are continuous in time. Such descriptions would involve an equation for the rate of change of the mass of A, another for the rate of change of the center of mass (CM) of A, one more for the rate of change of the CM velocity of A, and a last one governing how the shape of A changes. We also find a similar set of equations for B. Once they become connected, we need a single description that is continuous in time, but we do not completely discard the earlier descriptions of A and B. Instead, we think of the single description of A+n+B as being obtained by introducing one more set of equations for the neck n (which will eventually become the persistent structure C) and impose constraints on these equations. For example, $m_{A}+m_{B}+m_{n}$ must now be approximately conserved, and similarly for the momentum. In this merging stage, it is actually inconvenient to use only one set of equations for C=A+B, because too many things are changing simultaneously. It is convenient to use one set of equations for C only after A and B are completely absorbed by the neck.

## 4. Results and Discussion

### 4.1. Data

The daily prices of 671 Taiwan Stock Exchange (TWSE) stocks from 1 April 2018 to 30 September 2020 (Figure 10b), 530 Singapore Exchange (SGX) stocks from 31 August 2019 to 30 April 2021 (Figure 10a), and 504 component stocks of the S&P 500 from 1 June 2019 to 31 December 2020 (Figure 10c) were downloaded from Yahoo! Finance using Python’s pandas_datareader module. We then post-processed the financial time series as follows. First, “NaNs” were replaced with “0s”. Moreover, if the time series contains more than 50% “0s”, we remove this ticker symbol from the list. For the remaining stocks, we applied standardization, and also computed their returns. For SGX, some delisted stocks were downloaded manually from the investing.com website. Similarly, a few S&P 500 component stocks changed during the period of study, and so we downloaded both new and old component stocks from the investing.com website.

### 4.2. Methods

First, we identified four periods, each with a market crash (on TWSE, SGX, or S&P 500) in the middle, as shown in Table 2. We then computed the Pearson cross correlations
(3)$$C_{ij}=\frac{\sum_{t=1}^{N}\left(r_{i,t}-{\bar{r}}_{i}\right)\left(r_{j,t}-{\bar{r}}_{j}\right)}{\sqrt{\sum_{t=1}^{N}{\left(r_{i,t}-{\bar{r}}_{i}\right)}^{2}}\sqrt{\sum_{t=1}^{N}{\left(r_{j,t}-{\bar{r}}_{j}\right)}^{2}}}$$
of the daily returns $r_{i,t}$ and $r_{j,t}$ within a six-month time window with N+1 trading days, which we advanced one week at a time. Here, ${\bar{r}}_{i}$ and ${\bar{r}}_{j}$ are the average returns of stocks i and j within each six-month time window. For each time window, we further convert the pairwise cross correlations $C_{ij}$ into pairwise ultrametric distances $0\leq d_{ij}=\sqrt{2\left(1-C_{ij}\right)}\leq 2$.

Next, for a given market crash and each of its distance matrices, we perform the TDA filtration process by varying the filtration parameter ϵ. Two stocks, i and j, are linked if $d_{ij}\leq \epsilon$. Therefore, for a given time window at filtration parameter ϵ, we constructed an adjacency matrix $A_{ij}$ whose matrix elements are $A_{ij}=1$ if $d_{ij}\leq \epsilon$, and $A_{ij}=0$ otherwise. Using the adjacency matrix, we then computed the degree matrix whose diagonal elements are
(4)$$K_{ii}=k_{i}=\underset{j\neq i}{\sum }A_{ij},$$
and whose off-diagonal elements are $K_{ij}=0$. Finally, we constructed the graph Laplacian L(ϵ)=K−A to obtain its eigenvalues and eigenvectors. Over a judicious choice of filtration parameters, ϵ=0.5, 0.8, 1.0, 1.2, 1.4, 1.6, 1.8, 2.0, we then visualize the spectral sequence $\left\{\lambda_{i}\left(\epsilon \right)\right\}$ for each time window, but analyzed the spectral sequences and Fiedler vectors for the selected time windows.

### 4.3. March 2020 TWSE Crash

We start by analyzing the spectral sequences for the March 2020 TWSE crash, which was said to be caused by the start of the COVID-19 pandemic [39,40]. The complete series of spectral sequences can be found in Appendix A. Here, we show in Figure 11 the spectral sequences for only four time windows: (1) 1 August 2019–31 January 2020, (2) 22 September 2019–22 March 2020, (3) 15 October 2019–15 April 2020, and (4) 1 April 2020–30 September 2020. The first time window is before the March 2020 TWSE crash, while the fourth time window is after the March 2020 TWSE crash. The March 2020 TWSE crash occurred at the end of the second time window, and in the middle of the third time window.

From Appendix A, we see that the spectral sequences changed very rapidly from the 15 September 2019–15 March 2020 time window (that just missed the March 2020 TWSE crash) to the 22 September 2019–22 March 2020 time window (that first that included the March 2020 TWSE crash). For the first seven time windows that do not include the March 2020 TWSE crash, their spectral sequences resemble that of the 1 August 2019–31 January 2020 time window shown in Figure 11a, which in turn resembles that of a single cluster of points shown in Figure 7 of Section 3.3.1. For the 21 time windows overlapping the March 2020 TWSE crash, their spectral sequences are similar to those of the 22 September 2019–22 March 2020 (Figure 11b) and 15 October 2019–15 April 2020 (Figure 11c) time windows. These spectral sequences bear similarities to those shown in Figure 8 of Section 3.3.2, Appendix A, where we find prominent persistent gaps near the ends of the spectral sequences. Finally, the last four time windows shown in Appendix A have spectral sequences similar to the first seven time windows, as well as Figure 11d, suggesting that the TWSE had recovered from the March 2020 crash. These observations are consistent with the suspicion by econophysicists that a market crash is a critical transition. They also suggest that the March 2020 TWSE crash was short, lasting only for the first two weeks of March 2020 (indeed this is the time to go from the TAIEX high of 11,321 on 1 March 2020 to the low of 9234 on 15 March 2020), and seen as the “V”-shape feature in Figure 10b. They also agree with the picture of a market crash being the result of the fragmentation of a giant cluster.

For the first seven time windows and the last four time windows, we find a narrow band of eigenvalues (0≤λ<10) for the smallest filtration parameter ϵ=0.5. This tells us that at this scale, most of the clusters are small, and therefore the total number of clusters is comparable to the total number of stocks on the TWSE. The narrow bandwidth at ϵ=0.5 is consistent with only localized random walks on small, disconnected components. On the other hand, for the 21 time windows whose spectral sequences show prominent gaps, there is a broad band of eigenvalues (0≤λ<300) for ϵ=0.5. This suggests that at this scale, there is a broad distribution of cluster sizes, including a few strongly correlated ones with up to 300 stocks during the market crash. For these time windows, the broad bandwidth at ϵ=0.5 is consistent with the delocalization of random walkers on larger connected components.

Moreover, before and after the March 2020 TWSE crash, $\lambda_{1}$ rose rapidly at ϵ≈1.3, whereas during the market crash, $\lambda_{1}$’s rapid rise only began at ϵ≈1.7. This delay in the rapid rise of $\lambda_{1}$ suggests that during the market crash, the gap in correlations between clusters is 30–40% larger than the standard deviation in the continuous distribution of correlations within the giant cluster prior to its fragmentation. Indeed, the persistent gap is most pronounced at ϵ=1.6, although in some time windows, this persistent gap can also be observed at ϵ=1.4 or ϵ=1.8. Finally, as we elucidate the picture of March 2020 TWSE crash as a strongly correlated giant cluster fragmenting into a few strongly correlated clusters, let us clarify that this need not involve all stocks. Unaffected stocks then form a noisy background, whose effect is to obfuscate the persistent gap. Based on our analysis in Appendix A, the persistent gap can nevertheless be identified from the late rise in $\lambda_{1}$. This is indeed what we observed.

From Section 3.4, we understand that when two clusters become first connected by a thin neck, components of the two clusters have opposite signs in ${\vec{u}}_{1}$, while components of the neck have significantly smaller or zero weights. However, as the neck becomes thicker with increasing ϵ, components become distributed about zero, and few members of the two clusters remain distinguishable. With these in mind, let us start our eigenvector analyses with the time window 1 August 2019–31 January 2020, which was before the March 2020 COVID-19 crash. From Appendix A, we see that $\lambda_{0}=0$ is non-degenerate for 1.2≤ϵ≤2.0, and $\lambda_{1}$ changes most sharply between ϵ=1.4 and ϵ=1.6. Let us therefore examine the Fiedler vector ${\vec{u}}_{1}$ for 1.2≤ϵ≤1.8. For ϵ=1.2, $\lambda_{1}=19.538$, most of the Fiedler components have an absolute value of around $10^{-3}$, except for one component whose value is 0.991. To check for non-overlapping distributions that represent Fiedler components from the two clusters, we therefore limit ourselves to bins between −0.005 and +0.005 to plot high-resolution histograms in Figure 12. The distributions of Fiedler components at different filtration parameters for this time window are indeed consistent with there being just one giant cluster in the market before the crash.

While the picture for ϵ=1.2 is not clear in Figure 12a, for ϵ=1.4 it is clear from Figure 12b that most of the stocks (with negative components) were organized into a giant cluster, while most of the rest (with positive components) were organized into a minor cluster (shown in Appendix A). As expected, when ϵ≥1.6 (Figure 12c,d), the distribution of Fiedler components become unimodal, and centered around zero. Nevertheless, in Figure 12d we see that remnants of the two clusters are still visible at ϵ=1.8, with 76 components larger than 0.001, and 79 components less than −0.001. The cluster with 79 components is shown in Appendix A.

Moving on to the 22 September 2019–22 March 2020 time window (Appendix A, which includes the first week of the crash), $\lambda_{0}=0$ is again non-degenerate for 1.2≤ϵ≤2.0, and $\lambda_{1}$ now changes most sharply between ϵ=1.6 and ϵ=1.8. At ϵ=1.2 and ϵ=1.4, there is a noticeable gap in the distribution of Fiedler components, between those that are negative, and those that are positive. The smaller of these groups are shown in Appendix A. However, we must be careful interpreting all of these components as part of the minor cluster, as we can see from Figure 12e,f that some of these components are close to zero, and might be part of the neck instead. Since $\lambda_{1}$ changes most rapidly between ϵ=1.6 and ϵ=1.8, we therefore expect the distribution of Fiedler components at ϵ=1.6 to be similar to those at ϵ=1.4. Indeed, two clusters can be identified, but there is now a larger neck with close-to-zero components. Based on the small gap at around −0.001 (Figure 12g), we identified members of the minor cluster, as shown in Appendix A. Finally, at ϵ=1.8, most of the components have become zero, suggesting that the neck (566 stocks) has grown to dominate the two clusters. Roughly 100 stocks of the major cluster and 5 stocks of the minor cluster remain distinguishable, as we can see from Figure 12h. As we can see from Appendix A, a smaller minor cluster identified at a given ϵ is almost perfectly embedded in the larger minor cluster identified at the preceding or succeeding ϵ. This self-consistency at different length scales helps us to reliably identify the two clusters within a given time window.

Moreover, from the spectral sequence of this time window, we see that there is a pair of nearly degenerate eigenvalues $\lambda_{1}=38.693$ and $\lambda_{2}=46.046$ at ϵ=1.4. This pair of eigenvalues came from a larger group of nearly degenerate eigenvalues at ϵ=1.2. Unlike the situation shown in Figure 9, where two smaller clusters merge first before merging later with a third cluster, having two small eigenvalues suggests the formation of two thin necks, as shown in Figure 9. When we examine the ratio $r_{i}=u_{1,i}/u_{2,i}$ at ϵ=1.4 (Figure 13(left)), we find a group of 622 components that can be distinguished from the remaining 49 components. These 622 components contain the major cluster, whose components have ratios around $r_{i}=1.4$. Of the 49 components that do not belong to the major cluster, 14 has $-0.5<r_{i}<0.5$, and are the most likely candidate for the bridging cluster shown in Section 3.4.3. From Section 3.4.3, we also understood that components with very large absolute ratios $r_{i}=u_{1,i}/u_{2,i}$ are members of the necks. Specifically, those with $r_{i}<-2$ have been identified alongside the minor cluster at ϵ=1.4 and ϵ=1.6 in Appendix A. The rest are likely members of the neck that link the major cluster to the bridging cluster, as shown schematically in Figure 14.

Next, let us move on to the 15 October 2019–15 April 2020 time window (Appendix A), which covers both weeks of the crash. In this time window, $\lambda_{0}=0$ is non-degenerate over 1.2≤ϵ≤2.0, while $\lambda_{1}$ changes most rapidly between ϵ=1.6 and ϵ=1.8. At ϵ=1.2, $\lambda_{1}=2.979$, 651 of the stocks are in the major cluster, while the minor cluster contains the 4 stocks shown in Appendix A. At this filtration parameter, there are no components close to zero. When we go to ϵ=1.4, $\lambda_{1}=25.758$, we find three sub-distributions of components. The first sub-distribution, containing 625 components, represents the major cluster. The second sub-distribution, containing 18 components close to zero, represents either the neck, or a bridging cluster. The third sub-distribution, containing the 12 components shown in Appendix A, represents the minor cluster. At ϵ=1.6, $\lambda_{1}=106.97$, the sub-distribution centered about zero becomes well defined. Nevertheless, there are 5 components remaining in the minor cluster, as shown in Appendix A. Here, we see that the minor cluster for ϵ=1.6 is completely embedded in the minor cluster for ϵ=1.4. Finally, when ϵ=1.8, $\lambda_{1}=563$, nearly all components are close to zero, but remnants of the major cluster (136 components) and minor cluster (6 components) can still be seen.

Just like the 22 September 2019–22 March 2020 time window, in this time window there is also a pair of nearly degenerate eigenvalues $\lambda_{1}$ and $\lambda_{2}$. Unlike for the 22 September 2019–22 March 2020 time window, where the near degeneracy occurs only at ϵ=1.4, in the 15 October 2019–15 April 2020 time window this near degeneracy can be seen for 1.4≤ϵ≤1.8. At ϵ=1.4, $\lambda_{1}=25.758,$ $\lambda_{2}=34.013,$ we see from Figure 13(right) 621 narrowly distributed components associated with the major cluster in $r_{i}=u_{1,i}/u_{2,i}$. Of the remaining ratios, 11 have absolute values close to zero, and may be associated with the bridging cluster, while if we set the threshold to $r_{i}>1.0$, we find 18 neck components. $\lambda_{1}$ and $\lambda_{2}$ are also quasi-degenerate at ϵ=1.6 and ϵ=1.8, but the bridge and neck components identified from these two filtration parameters are different from those identified at ϵ=1.4.

Finally, let us analyze Fiedler vectors in the 1 April 2020–30 September 2020 time window, which has no overlap with the March 2020 TWSE crash. As we can see from Figure 11d, $\lambda_{0}=0$ is non-degenerate for 1.2≤ϵ≤2.0, while $\lambda_{1}$ changes most rapidly between ϵ=1.4 and ϵ=1.6. At ϵ=1.2, $\lambda_{1}=4.882$, we see in Figure 12m that there is a single distribution of Fiedler components. When ϵ=1.4, $\lambda_{1}=110.37,$ we see from Figure 12n that there are now two sub-distributions of Fiedler components. The first represents a major cluster, while the second (shown in Appendix A), extending from zero to 0.005, probably includes both the neck and the minor cluster. When ϵ=1.6, the larger sub-distribution of Fiedler components is the one about zero (Figure 12o), even though the remnant sub-distribution associated with the major cluster is still sizeable. The sub-distribution of the minor cluster (shown in Appendix A) overlaps with that of the neck, making it difficult to isolate. Finally, at ϵ=1.8, we find a narrow sub-distribution of Fiedler components about zero (Figure 12p), and two weak sub-distributions away from zero. The latter represent remnants of the major and minor clusters.

### 4.4. September 2018 TWSE Mini-Crash

After the detailed analyses shown in Section 4.4, the natural question that comes to mind is how much of what we have found there is universal, i.e., applies to all market crashes, and how much of these are peculiar to the March 2020 TWSE crash. To answer this question, we repeated our spectral sequence and Fiedler vector analyses for two other market crashes. The first such crash is the September 2018 TWSE mini-crash in this section, so that we can ascertain universal features of market crashes over at least two crashes on the TWSE. The second such crash is the March 2020 SGX COVID-19 crash in Section 4.6, and also Section 4.6 so that we can confirm universal features of the same market crash (COVID-19 crash) at least across three different markets.

To do this, let us start with the gross features seen in the spectral sequences in Figure 15. From Appendix A, we see that the spectral sequences of the first three time windows and the last four time windows resemble that from a single cluster of data points, whose spectral sequence is characterized by the absence of persistent gaps. These suggest that the TWSE was in a *gapless* normal phase prior to the September 2018 mini-crash, and returned to the normal phase after the mini-crash. For time windows overlapping the mini-crash, the spectral sequences are characterized by persistent gaps at ϵ=1.4 and/or ϵ=1.6. The persistent gaps (appearing over a broad range of ϵ) for this mini-crash appear to be weaker than the ones seen for the COVID-19 crash, but they are qualitatively similar. Therefore, a gapped spectral sequence appears to be a universal feature associated with a *gapped* market crash phase, even though the strength of the gap may vary from crash to crash. A closely related (dilation) universal feature that we can identify from the spectral sequences of the two TWSE crashes is the filtration parameter value ϵ at which $\lambda_{1}$ changes most rapidly. For both crashes, $\lambda_{1}$ rises sharply between ϵ=1.2 and ϵ=1.4 in the normal phase, but is delayed till to between ϵ=1.6 and ϵ=1.8 in the crash phase. Moreover, during the mini-crash, which lasted about four months according to the number of spectral sequences with persistent gaps (agreeing with the “U”-shaped feature seen in Figure 10b), the most pronounced change occurred when we go from the 1 October 2018–28 February 2019 time window to the 8 October 2018–8 March 2019 time window, where the clear gap at ϵ=1.6 seen in the former completely disappeared in the latter. Therefore, unlike the COVID-19 crash, where the transition into the crash phase is sharp but not the transition out of the crash, the September 2018 TWSE mini-crash shows the opposite behavior, whereby the transition into the crash phase is not sharp, but the transition out of the crash is.

We also analyzed the Fiedler components at different ϵ over the four selected time windows associated with the September 2018 TWSE mini-crash. Their histograms are shown in Figure 16. As in Figure 13, we find the same evolution from bi-modal to unimodal distributions of Fiedler components as we increase ϵ. Just as for the March 2020 TWSE COVID-19 crash, the Fiedler vector points to the existence of a major cluster, comprising nearly all the stocks in the TWSE, and a minor cluster. The sub-distribution of Fiedler components associated with this minor cluster is weak in the time windows before and after the crash, and strong in the time windows overlapping the crash. However, unlike in the March 2020 COVID-19 crash, where $\lambda_{1}$ and $\lambda_{2}$ are quasi-degenerate at ϵ=1.4, suggesting the presence of two necks and necessitating the use of the ratio $r_{i}=u_{1,i}/u_{2,i}$ to identify a bridging cluster between the major and minor clusters, for the September 2018 mini-crash $\lambda_{1}$ and $\lambda_{2}$ are clearly different in the two time windows containing the crash, at ϵ=1.4. There is also a tri-modal feature in the distribution of Fiedler components at ϵ=1.6 (Figure 16o) in the time window right after the September 2018 mini-crash. We do not understand the meaning behind this feature, which is also absent from other time windows of the September 2018 mini-crash, and all time windows of the March 2020 COVID-19 crash.

### 4.5. March 2020 SGX Crash

For the SGX, we computed spectral sequences for 69 time windows in total, and show these as Appendix A. We included this many time windows for the SGX, because the COVID-19 crash on this market has a long “U”-shape (see Figure 10a), unlike the short “V”-shaped COVID-19 crash on the TWSE (see Figure 10b). This difference is due to the different COVID-19 pandemic trajectories in the two regions: where Taiwan managed to keep COVID-19 at bay over nearly the whole of 2020 (hence the “V”-shaped crash), Singapore succumbed to the pandemic and had to enact strict population health measures starting April 2020 (hence the “U”-shaped crash). To avoid missing the actual recovery from the crash, we made sure that our spectral sequences cover the whole “U”-shaped period. Out of these time windows, the first six and the last 39 spectral sequences are reminiscent of the spectral sequence of a single cluster. This finding on the SGX further supports our universality hypothesis that the gapless normal phase of a stock market consists of a single undifferentiated cluster, based on our findings on the TWSE in Section 4.4 and Section 4.5. The remaining 24 spectral sequences were found to be gapped, suggesting that these 24 time windows overlapped with the March 2020 SGX COVID-19 crash. Based on the Straits Times Index (STI), the SGX reached a high on 9 February 2020, but according to the spectral sequences the crash only started on or after 8 March 2020, and surprisingly returned to normal on or after 15 March 2020. This tells us that the spectral sequence method is sensitive to the difference between the normal and crash phases of a stock market, and can therefore be used to time the start and end of crashes, instead of using the index value.

The dilation feature seen during the September 2018 and March 2020 TWSE crashes is even more pronounced during the March 2020 SGX crash. This supports our hypothesis that this dilation feature, which is closely associated with the opening of the spectral gap, is universal across markets. Just as for the TWSE, $\lambda_{1}$ changes most sharply between ϵ=1.4 and ϵ=1.6 in the normal phase, but between ϵ=1.6 and ϵ=1.8 in the crash phase of the SGX. In fact, in Figure 17c, $\lambda_{1}$ changes most sharply between ϵ=1.8 and ϵ=2.0. During the SGX COVID-19 crash, we found non-persistent gaps appearing at ϵ = 1.4, 1.6, and 1.8, which suggest that the size distribution of large persistent clusters is discrete, just like it was for TWSE. The main difference between the TWSE and the SGX, is the SGX having between 3 and 6 zero eigenvalues at ϵ=1.2.

Moving on, we showed the histograms of Fiedler components over the March 2020 SGX COVID-19 crash in Figure 18. For the (first row) 1 August 2019–31 December 2019 and (last row) 8 April 2020–8 October 2020 time windows, the distributions of Fiedler components are mostly unimodal, except for ϵ=1.6. This agrees with our observation for both crashes on the TWSE. For the (second row) 8 October 2019–8 March 2020, (third row) 8 November 2019–8 April 2020, (fourth) 22 February 2020–22 August 2020, and (fifth row) 8 March 2020–8 September 2020 time windows, the distributions of Fiedler components are bimodal, even up to ϵ=1.8. Again, this agrees with our observation for both crashes on the TWSE.

### 4.6. March 2020 S&P500 Crash

In addition to emerging markets such as TWSE and SGX, we also investigated the component stocks of S&P 500 from 1 June 2019 to 31 December 2020. Our target is again the COVID-19 crash, which occurred between 1 and 8 March 2020 in the S&P 500 according to the spectral sequences shown in Appendix A, compared to 1–15 March 2020 in TWSE and 8–15 March 2020 in SGX. Compared to the March 2020 TWSE crash (whose beginning was sharp, but whose ending was not) and the March 2020 SGX crash (whose beginning and ending were both sharp), the beginning and end of the March 2020 S&P 500 crash were both not sharp. In fact, according to conventional indicators, the S&P 500 attained a high of 3380 on 14 February 2020, and a low of 2304 on 20 March 2020.

As expected, the spectral sequence of the S&P 500 stocks is gapless in the normal phase, and gapped in the crash phase. This suggests strongly that the existence of a persistent spectral gap distinguishes the crash phase from the normal phase, whether or not the stock market is emerging or mature. The difference between the S&P 500 (measuring the mature US markets) and the TWSE/SGX is the extent of the persistent spectral gap (going into smaller length scales) in the spectral sequences of the S&P 500, compared to those in TWSE and SGX. For the S&P 500, there is also a persistent description in terms of two clusters. We suspect this is because of the significant fraction of S&P 500 component stocks are traded on National Association of Securities Dealers Automated Quotations (NASDAQ), while the remainder are traded on the New York Stock Exchange (NYSE).

Next, we show the histograms of Fiedler components over the February 2020 S&P 500 COVID-19 crash in Figure 19. In agreement with what we found in the SGX and TWSE, the distributions of Fiedler components go from bimodal to unimodal as we increase ϵ. This transition coincides with $\lambda_{1}$ changing most rapidly with ϵ.

## 5. Conclusions

In summary, we explained using a simple raindrop analogy the concept of persistent structures, and why they are useful as mesoscopic variables for describing the dynamics of complex systems (for example, market crashes) in terms of continuous and discontinuous changes. We then drew inspiration from the connection between (1) the symmetries [A,H]=0 of a quantum system, and (2) the block-diagonal structure of the Hamiltonian, leading ultimately to (3) the organization of energy eigenvalues into bands separated by band gaps, to approach the problem of overlapping communities obtained during the filtration process in TDA. Instead of trying to identify such communities in real space, we should therefore look for signatures of community structure in spectral space. For this work, the graph Laplacian L=K−A (A being the adjacency matrix, and K being the (diagonal) degree matrix) plays the role of the Hamiltonian.

To check its feasibility, we tested the spectral approach on a series of toy models, from a single cluster of data points to two or more well-defined clusters of data points, characterized by gaps of different length scales, in the absence or presence of a noisy background. We then introduced the spectral sequence as a novel tool to visualize how the Laplacian spectra of different ϵ change over increasing filtration parameter ϵ. For a single cluster of data points, the spectral sequence is gapless, whereas for multiple well-defined clusters the spectral sequence contain gaps that persist over a wide range of ϵ. Connecting the real-space TDA and the analysis of Laplacian spectra, we proposed for these persistent gaps to be used as signatures of persistent structures in the data. Spectral gaps that are persistent with respect to changes in length scale tend to be persistent with respect to changes in time, and are robust with respect to background noise. We also analyzed the Fiedler vector ${\vec{u}}_{1}$ associated with the first non-zero Laplacian eigenvalue $\lambda_{1}>0$, which is well-known to contain information on community structure in the data. In the case of two merging clusters, we confirmed earlier studies that their components have different signs in ${\vec{u}}_{1}$, but within each cluster, components have roughly the same value. We also developed a new understanding that components with significantly smaller (or even zero) absolute magnitudes are members of the neck, a structure that must be considered as distinct from the clusters it connects. We also understood for the first time how there can be near degeneracy between $\lambda_{1}>0$ and $\lambda_{2}\approx \lambda_{1}$, when three clusters are arranged in a linear configuration, with two necks forming roughly around the same length scales. Members of the bridging cluster can be distinguished from members of the two necks by examining the ratios of their components in ${\vec{u}}_{1}$ and ${\vec{u}}_{2}$.

Finally, we tested this spectral approach to unravel persistent structures on the daily prices of 671 stocks of the TWSE, 530 stocks of the SGX, and 504 component stocks of the S&P 500. Based on the toy model studies, we realized that this approach is ideal for analyzing topological and geometrical changes in stock markets when they crash (fragmentation), and also when they recover (agglomeration). Therefore, we identified two time windows (1 April 2018 to 30 April 2019, and 1 August 2019 to 30 September 2020) associated with two crashes on the TWSE, one time window (1 August 2019 to 30 April 2021) associated with the crash on the SGX, and one time window (1 June 2019 to 31 December 2020) associated with the crash on the S&P 500. We then computed the Pearson cross correlations $C_{ij}$ between stocks in six-month windows, converted $C_{ij}$ into pairwise distances $D_{ij}$ for the TDA filtration process, before sliding the time window one week at a time. We found universally across market crashes and stock markets that (1) the spectral sequence is gapless (absence of persistent or non-persistent gaps) when the time window is entirely within the normal phase, (2) the spectral sequence is gapped (presence of persistent gaps at large length scales) when the time window overlaps with the market crash phase, (3) the most rapid change in $\lambda_{1}$ is delayed in the crash phase relative to the normal phase, (4) in the normal phase, the distribution of Fiedler components is predominantly uni-modal, (5) in the crash phase, the distribution of Fiedler components change from bi-modal to uni-modal at the filtration parameter where $\lambda_{1}$ changes most rapidly. These are all results not previously known.

Together, our spectral analyses of toy models and real-world stock market data suggests that two clusters A and B do not become a single cluster AB the moment they are linked by a neck, but continue to retain their distinct identities until their members are completely absorbed by the growing neck. This can be summarized by the fusion process A+B→A+n+B→a+N+b→N(=C). Within this new perspective, n→N→N represents the thickening of the neck, while A→a and B→b represent the absorption of A and B by the neck. This ternary fusion picture is useful regardless of whether the fusion is a result of increasing the length scale during the filtration process, or a result of interactions that bring the two clusters closer to each other over time. In terms of this ternary fusion process, we can explain many mysteries observed in real-world data that cannot be explained using only binary fusion processes.

Finally, we explored only the graph Laplacian, its spectrum, and its eigenvectors in this paper. However, we know that the graph Laplacian is the simplest member of a hierarchy of Hodge Laplacians, which we plan to explore in our future works.

## Figures and Tables

**Figure 1 entropy-25-00846-f001:**
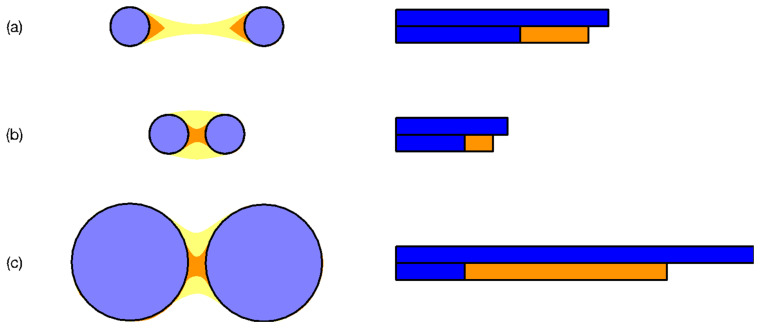
Three pairs of clusters at three increasing filtration parameters $\epsilon_{1}$ (no necks, communities shown in blue), $\epsilon_{2}$ (necks shown in orange), and $\epsilon_{3}$ (necks shown in yellow). For each pair of clusters, we also show the standard TDA barcode (blue bars, from ϵ=0 to the value of ϵ when the clusters become connected), and an extended barcode shown in orange where the original clusters remain distinguishable. In (**a**), the two small clusters remain distinguishable over a large range of filtration parameters. This is to be contrasted against (**b**), where the two clusters are the same sizes as those in (**a**), but are closer to each other. They are therefore distinguishable only over a small range of filtration parameters. Finally, in (**c**), we have two large clusters whose separation is the same as that in (**b**). However, because of their sizes, the two large clusters are distinguishable over a much larger range of filtration parameters.

**Figure 2 entropy-25-00846-f002:**
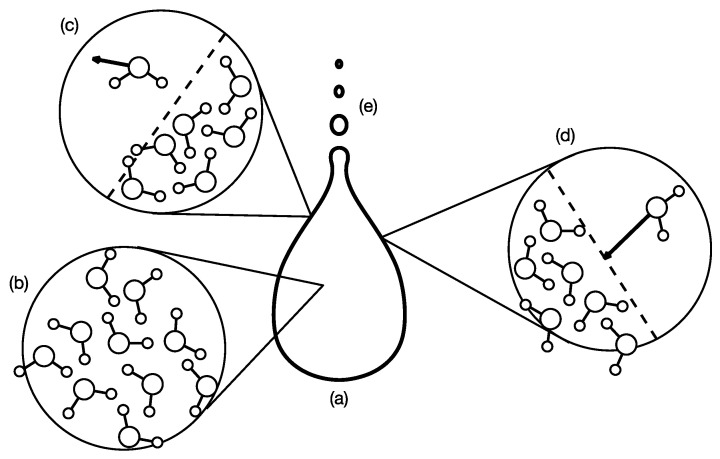
(**a**) A macroscopic raindrop with its characteristic teardrop shape falling through air. (**b**) The raindrop consists of a large number of microscopic water molecules whose relative positions are always changing. (**c**) Every now and then, a water molecule will escape from the surface of the raindrop (shown as dashed line). (**d**) Sometimes, the raindrop (whose surface is shown as a dashed line) can also absorb a water molecule from the air around it. (**e**) If the raindrop falls too fast, its surface will become unstable, and the trailing end of the raindrop may breakup into smaller droplets.

**Figure 3 entropy-25-00846-f003:**
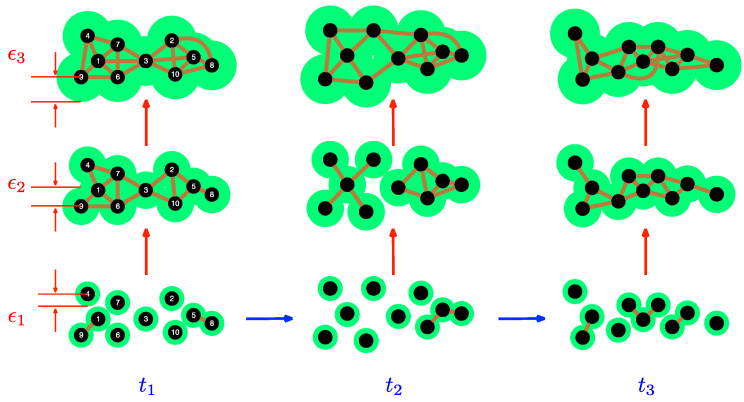
The simplicial complexes obtained by the TDA filtration process for a set of ten data points at three different scales $\epsilon_{1}<\epsilon_{2}<\epsilon_{3}$ (the sizes of the green disks), at three different times $t_{1}<t_{2}<t_{3}$. In this figure, two data points ${\vec{r}}_{i}$ and ${\vec{r}}_{j}$ are connected, if $\left|{\vec{r}}_{i}-{\vec{r}}_{j}\right|\leq \epsilon$.

**Figure 4 entropy-25-00846-f004:**
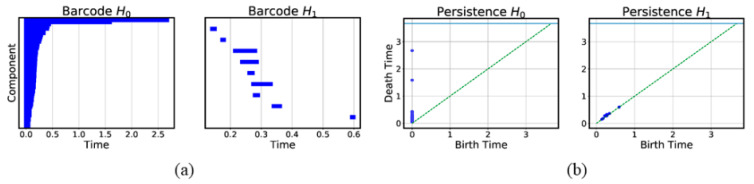
(**a**) The barcodes of the 0-dimensional homology group $H_{0}$ and 1-dimensional homology group $H_{1}$ for an artificial data set with 50 data points (the same one as in the 1st Figure in Section 3.3.2) undergoing the filtration process. (**b**) The persistence diagrams of the 0-dimensional and 1-dimensional components emerging from the filtration process.

**Figure 5 entropy-25-00846-f005:**
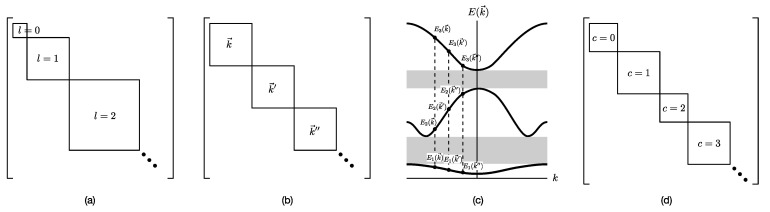
(**a**) The matrix representation of the angular momentum operator $L^{2}$ is block-diagonal when it is written in the basis of eigenstates of $L_{z}$. Within a diagonal block, all states |l m〉 have eigenvalue $l\left(l+1\right)\hbar^{2}$ for $L^{2}$, and eigenvalue mℏ, m=−l,…,0,…,l for $L_{z}$. (**b**) For the Hamiltonian matrix of a crystal in momentum space, we find one diagonal block associated with each wave vector $\vec{k}$. (**c**) When we diagonalize the block-diagonal matrix shown in (**b**), we find the eigenvalues organized into bands $E_{n}\left(\vec{k}\right)$ separated by band gaps (gray). (**d**) For a network with community structure, the adjacency matrix A or the Laplacian matrix L is also block diagonal, with each block associated with a different community with community index c.

**Figure 6 entropy-25-00846-f006:**
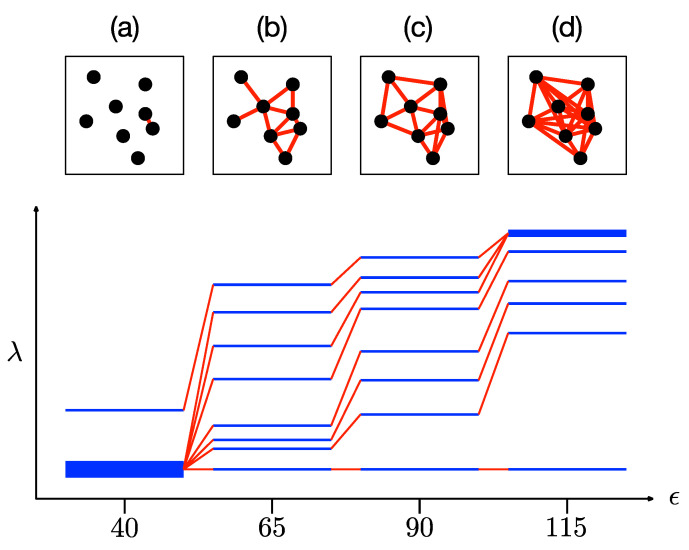
A set of eight data points going through the filtration process, and the resulting Laplacian spectra (blue horizontal lines) for filtration parameters (**a**) ϵ=40, (**b**) ϵ=65, (**c**) λ=90, and (**d**) λ=115. Thin blue lines tell us that the eigenvalues are nondegenerate, whereas thick blue lines indicate that the eigenvalue is degenerate. We use red lines to connect $\lambda_{n}\left(\epsilon \right)$ to $\lambda_{n}\left(\epsilon^{\prime }\right)$, for successive filtration parameters $\epsilon^{\prime }>\epsilon$.

**Figure 7 entropy-25-00846-f007:**
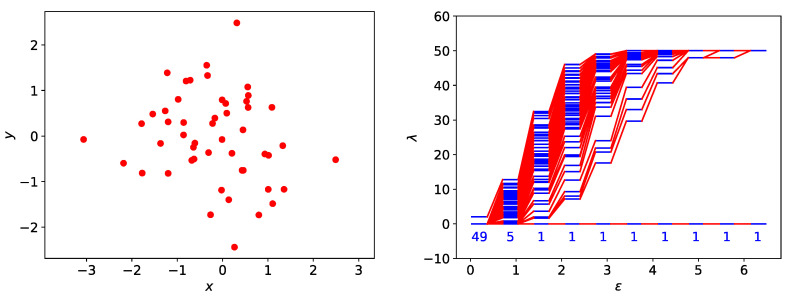
(**left**) A cluster of 50 data points sampled from a two-dimensional normal distribution $p\left(x_{1},x_{2}\right)=\frac{1}{2\pi \left|\Sigma \right|}\exp\left[-\frac{1}{2}{\left(\vec{x}-\vec{\mu }\right)}^{T}\Sigma^{-1}\left(\vec{x}-\vec{\mu }\right)\right],$ where $\vec{\mu }=\left(\mu_{1},\mu_{2}\right)=\left(0,\;0\right)$ and Σ is a diagonal covariance matrix with diagonal matrix elements $\sigma_{11}^{2}=1$ and $\sigma_{22}^{2}=1$. (**right**) The spectral sequence (i.e., the distribution of eigenvalues $\left\{\lambda_{n}\left(\epsilon \right)\right\}$ of the Laplacian matrix L at different filtration parameter ϵ) of this cluster. In this figure, the numbers of zero eigenvalues for different ϵ are indicated below λ=0.

**Figure 8 entropy-25-00846-f008:**
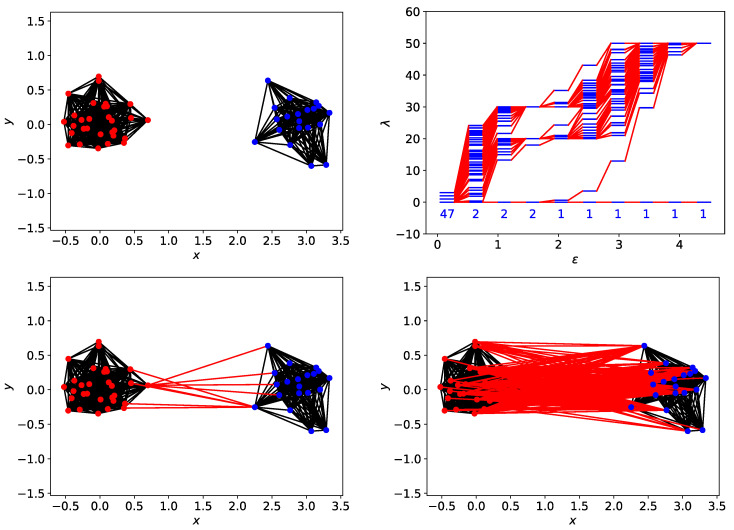
(**top right**) The spectral sequence for two clusters, one with 30 red data points, the other with 20 blue data points. In this figure, the persistent spectral gap that corresponds to this spatially persistent two-cluster structure is shaded yellow. (**top left**) The simplicial complex of the two clusters at ϵ=0.9855, which consists of the two nearly complete networks that are not connected. (**bottom left**) The simplicial complex of the two clusters at ϵ=1.9254, showing how the red cluster is connected to the blue cluster by 9 links, between 5 red nodes and 5 blue nodes. (**bottom right**) The simplicial complex of the two clusters at ϵ=2.8653. At this length scale, the two clusters are connected by 261 links. In these figures, intra-cluster links are black, while inter-cluster links are red.

**Figure 9 entropy-25-00846-f009:**
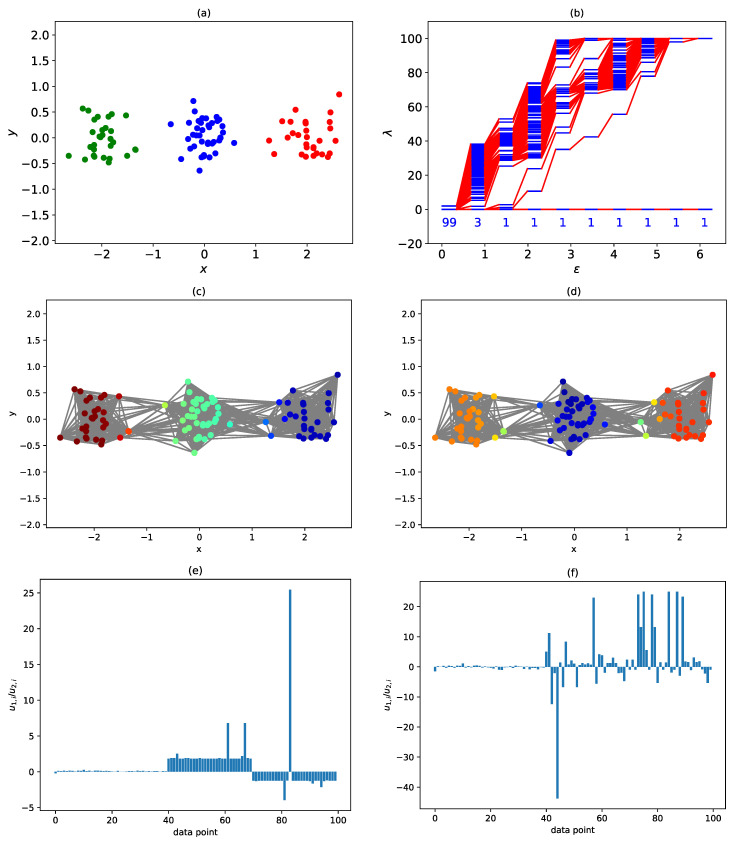
(**a**) Three clusters of data points arranged in a straight line. The blue cluster contains 40 data points, while the red and green clusters each contain 30 data points. (**b**) The spectral sequence of the three clusters of data points. Note the sudden change from a three-cluster description at ϵ=0.668 to a one-cluster description at ϵ=1.328. Note also the pair of small, closely spaced eigenvalues $\lambda_{1}=1.067$ and $\lambda_{2}=2.776$ at ϵ=1.328. (**c**) The data points are colored according to their components in ${\vec{u}}_{1}$, the eigenvector associated with $\lambda_{1}$. A similar example was shown by Servedio et al. in Refs. [37,38]. (**d**) The data points are colored according to their components in ${\vec{u}}_{2}$, the eigenvector associated with $\lambda_{2}$. (**e**) Ratio of components in ${\vec{u}}_{1}$ to components in ${\vec{u}}_{2}$ when ϵ=1.328. (**f**) Ratio of components in ${\vec{u}}_{1}$ to components in ${\vec{u}}_{2}$ when ϵ=1.987.

**Figure 10 entropy-25-00846-f010:**
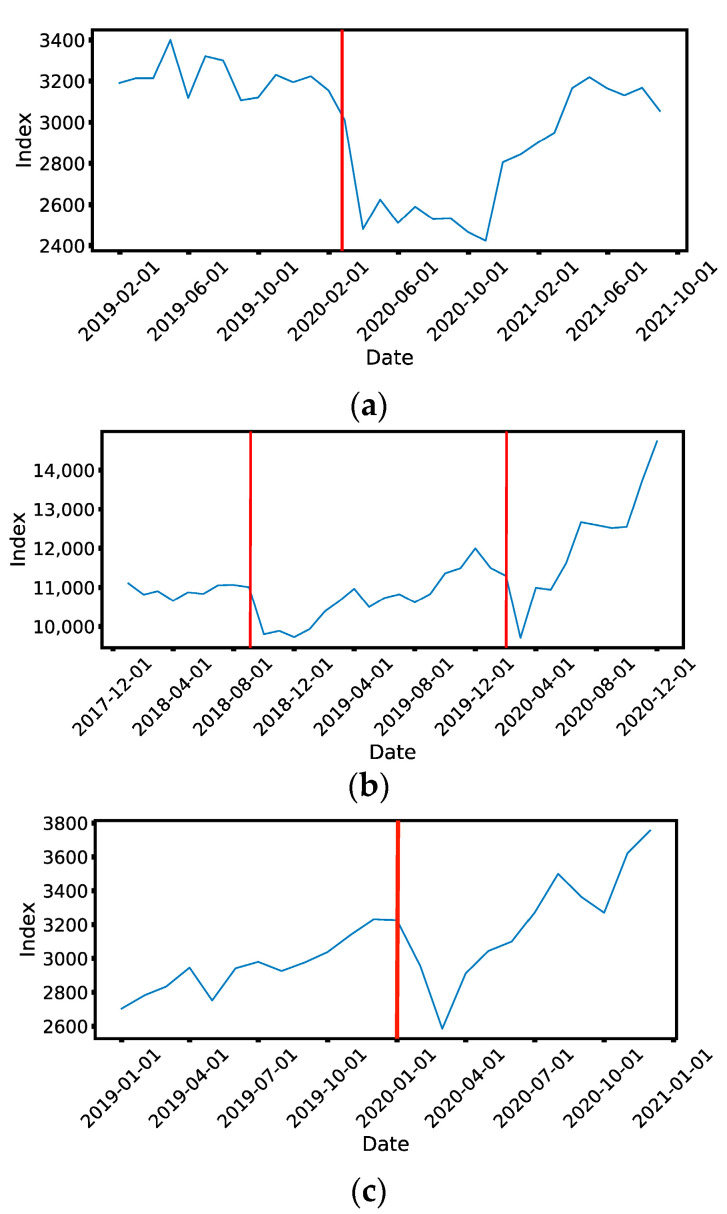
Monthly values of (**a**) the Straits Times Index (STI) of the SGX, (**b**) the Taiwan Capitalization Weighted Stock Index (TAIEX) of the TWSE, and (**c**) the Standard & Poor’s 500 (S&P 500) between 1 January 2019 to 31 August 2021, 1 January 2018 to 31 December 2020, and 1 June 2019 to 31 December 2020 respectively. We are specifically interested in two market crashes (Sep 2018 and March 2020) on the TWSE, one market crash (Mar 2020) on the SGX, and one market crash (Mar 2020) for the S&P 500. In these figures, these are shown as red vertical lines.

**Figure 11 entropy-25-00846-f011:**
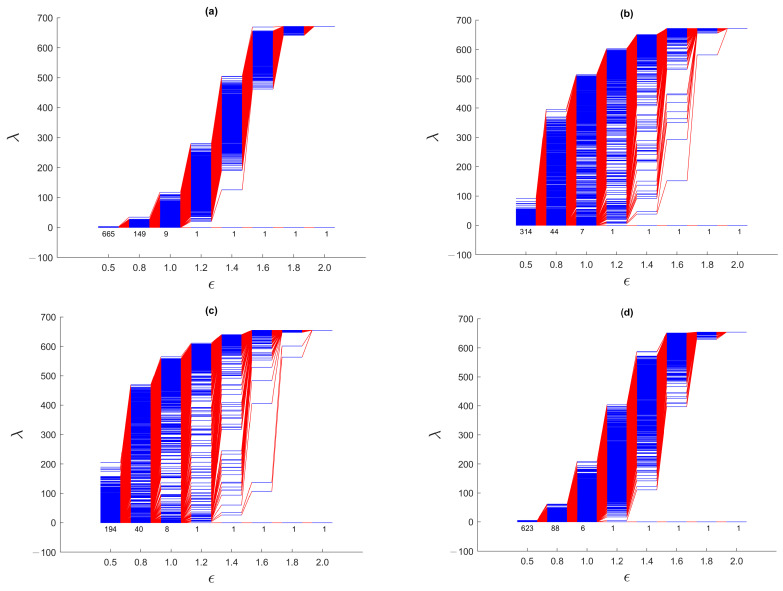
The spectral sequences of the TWSE for ϵ=0.5, 0.8, 1.0, 1.2, 1.4, 1.6, 1.8, 2.0 over the six-month time windows: (**a**) 1 August 2019–31 January 2020 (671 stocks), (**b**) 22 September 2019–22 March 2020 (671 stocks), (**c**) 15 October 2019–15 April 2020 (655 stocks), and (**d**) 1 April 2020–30 September 2020 (654 stocks). During the March 2020 TWSE crash, the TAIEX fell from a high of 11,321 on 1 March 2020 to a low of 9234 on 15 March 2020.

**Figure 12 entropy-25-00846-f012:**
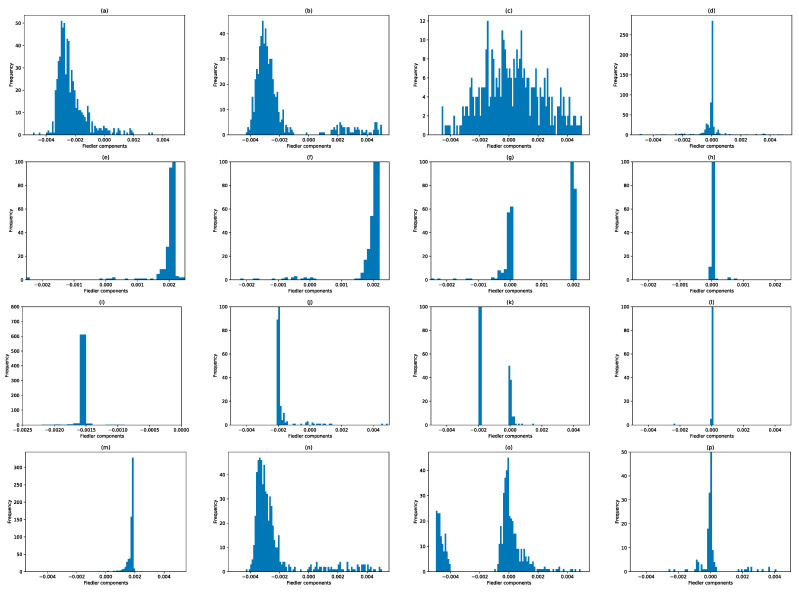
Distribution of Fiedler components at different filtration parameters: (first row) (**a**) ϵ=1.2, (**b**) ϵ=1.4, (**c**) ϵ=1.6, and (**d**) ϵ=1.8 for the 1 August 2019–31 January 2020 time window; (second row) (**e**) ϵ=1.2, (**f**) ϵ=1.4, (**g**) ϵ=1.6, and (**h**) ϵ=1.8 for the 22 September 2019–22 March 2020 time window; (third row) (**i**) ϵ=1.2, (**j**) ϵ=1.4, (**k**) ϵ=1.6, and (**l**) ϵ=1.8 for the 15 October 2019–15 April 2020 time window, and (fourth row) (**m**) ϵ=1.2, (**n**) ϵ=1.4, (**o**) ϵ=1.6, and (**p**) ϵ=1.8 for the 1 April 2020–30 September 2020 time window.

**Figure 13 entropy-25-00846-f013:**
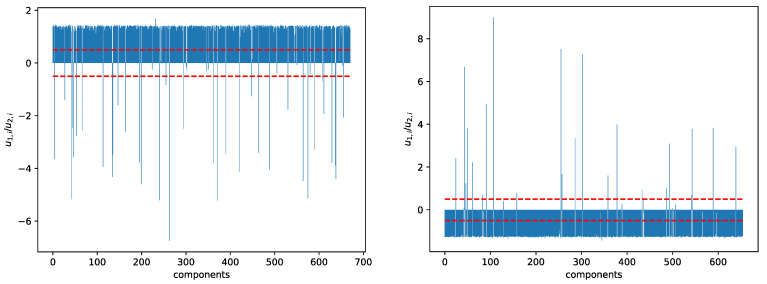
Bar plot of the ratio $u_{1,i}/u_{2,i}$ of components of the eigenvectors ${\vec{u}}_{1}$ and ${\vec{u}}_{2}$ associated with the smallest non-trivial eigenvalues $\lambda_{1}$ and $\lambda_{2}$ of the graph Laplacian obtained at filtration parameter ϵ=1.4, in (**left**) the 22 September 2019–22 March 2020 time window, and (**right**) the 15 October 2019–15 April 2020 time window. In this figure, components between the two red dashed lines are likely to be members of a bridging cluster.

**Figure 14 entropy-25-00846-f014:**
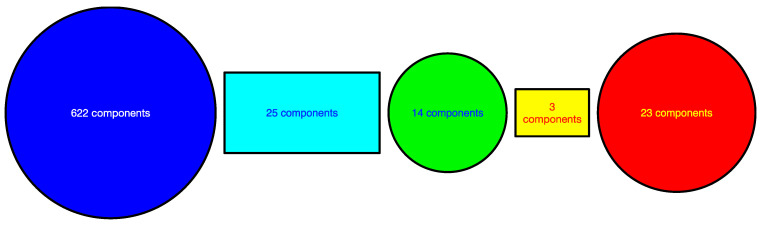
Schematic figure showing the major cluster (blue, 622 components) being linked to the minor cluster (red, 23 components) through a bridging cluster (green, 14 components) in the 22 September 2019–22 March 2020 time window. In the cyan neck between the blue and green clusters, there are 25 components. In the yellow neck between the green and red clusters, there are three components.

**Figure 15 entropy-25-00846-f015:**
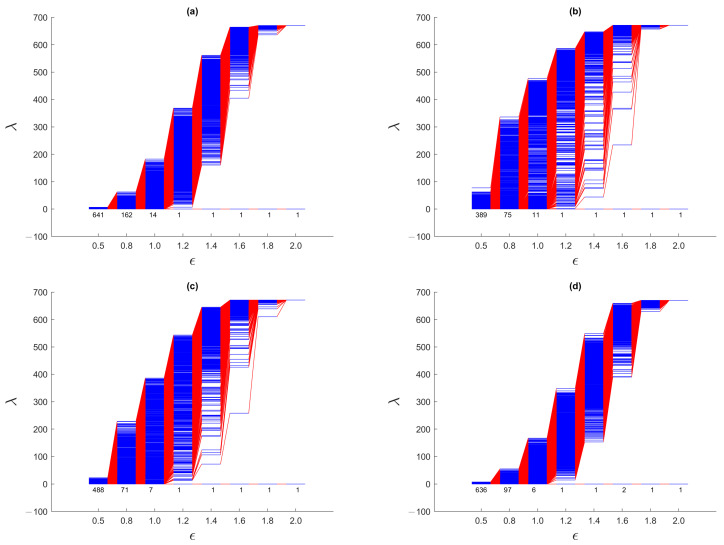
The spectral sequences of the TWSE for ϵ=0.5, 0.8, 1.0, 1.2, 1.4, 1.6, 1.8, 2.0 over the six-month time windows: (**a**) 8 April 2018–8 October 2018, (**b**) 22 August 2018–22 February 2019, (**c**) 1 October 2018–15 April 2019, and (**d**) 1 November 2018–30 April 2019. During the September 2018 TWSE mini-crash, the TAIEX fell from a high of 11,006 on 16 September 2018 to a low of 9489 on 21 October 2018. The TAIEX remained low, reaching 9382 on 30 December 2018, before it started rising again.

**Figure 16 entropy-25-00846-f016:**
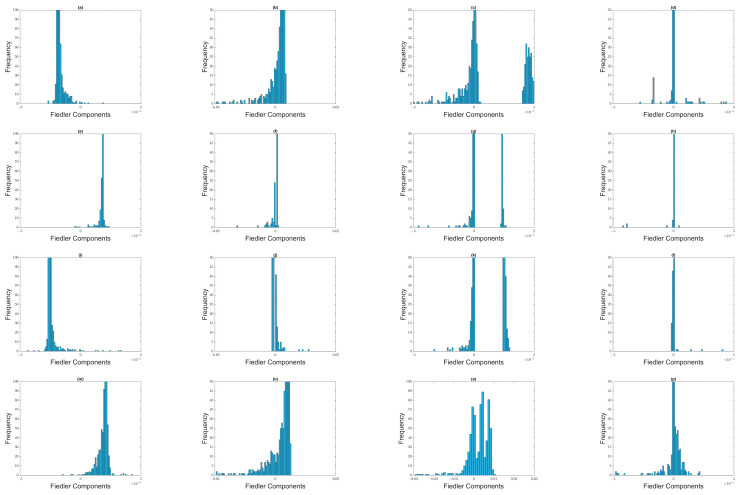
Distribution of Fiedler components at different filtration parameters: (first row) (**a**) ϵ=1.2, (**b**) ϵ=1.4, (**c**) ϵ=1.6, and (**d**) ϵ=1.8 for the 8 April 2018–8 October 2018 time window; (second row) (**e**) ϵ=1.2, (**f**) ϵ=1.4, (**g**) ϵ=1.6, and (**h**) ϵ=1.8 for the 22 August 2018–22 February 2019 time window; (third row) (**i**) ϵ=1.2, (**j**) ϵ=1.4, (**k**) ϵ=1.6, and (**l**) ϵ=1.8 for the 1 October 2018–1 March 2019 time window; and (fourth row) (**m**) ϵ=1.2, (**n**) ϵ=1.4, (**o**) ϵ=1.6, and (**p**) ϵ=1.8 for the 1 November 2018–1 April 2019 time window.

**Figure 17 entropy-25-00846-f017:**
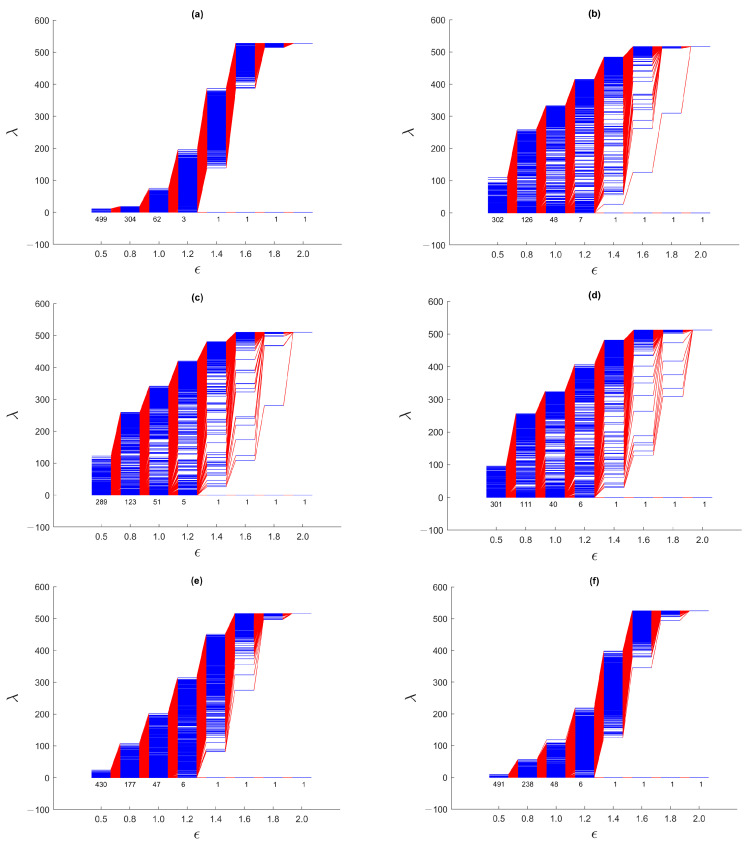
The spectral sequences of the TWSE for ϵ=0.5, 0.8, 1.0, 1.2, 1.4, 1.6, 1.8, 2.0 over the six-month time windows: (**a**) 1 August 2019–31 December 2019, (**b**) 8 October 2019–8 March 2020, (**c**) 8 November 2019–8 April 2020, and (**d**) 22 February 2020–22 August 2020, and (**e**) 8 March 2020–8 September 2020, and (**f**) 8 April 2020–8 October 2020. During the March 2020 STI crash, the STI fell from a high of 3220 on 9 February 2020 to a low of 2410 on 15 March 2020.

**Figure 18 entropy-25-00846-f018:**
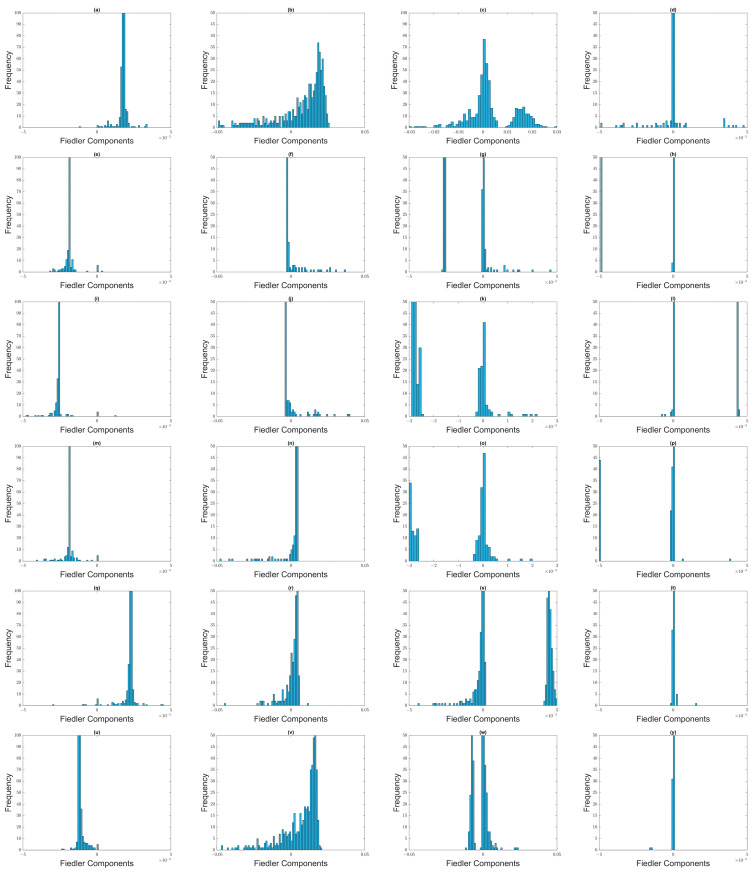
Distribution of Fiedler components at different filtration parameters: (first row) (**a**) ϵ=1.2, (**b**) ϵ=1.4, (**c**) ϵ=1.6, and (**d**) ϵ=1.8 for the 1 August 2019–1 January 2020 time window; (second row) (**e**) ϵ=1.2, (**f**) ϵ=1.4, (**g**) ϵ=1.6, and (**h**) ϵ=1.8 for the 8 October 2019–8 April 2020 time window; (third row) (**i**) ϵ=1.2, (**j**) ϵ=1.4, (**k**) ϵ=1.6, and (**l**) ϵ=1.8 for the 8 November 2019–8 May 2020 time window; (fourth row) (**m**) ϵ=1.2, (**n**) ϵ=1.4, (**o**) ϵ=1.6, and (**p**) ϵ=1.8 for the 22 February 2020–22 August 2020 time window; (fifth row) (**q**) ϵ=1.2, (**r**) ϵ=1.4, (**s**) ϵ=1.6, and (**t**) ϵ=1.8 for the 8 March 2020–8 September 2020 time window; and (sixth row) (**u**) ϵ=1.2, (**v**) ϵ=1.4, (**w**) ϵ=1.6, and (**y**) ϵ=1.8 for the 8 April 2020–8 October 2020 time window.

**Figure 19 entropy-25-00846-f019:**
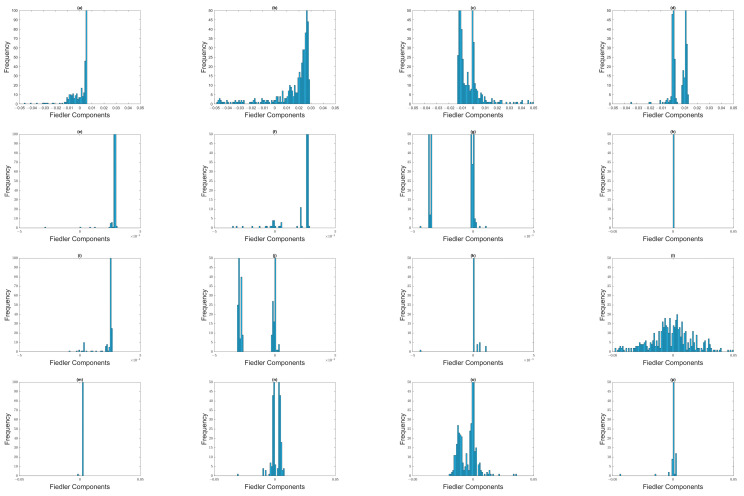
Distribution of Fiedler components at different filtration parameters: (first row) (**a**) ϵ=1.2, (**b**) ϵ=1.4, (**c**) ϵ=1.6, and (**d**) ϵ=1.8 for the 1 June 2019–30 November 2019 time window; (second row) (**e**) ϵ=1.2, (**f**) ϵ=1.4, (**g**) ϵ=1.6, and (**h**) ϵ=1.8 for the 08 September 2019–08 March 2020 time window; (third row) (**i**) ϵ=1.2, (**j**) ϵ=1.4, (**k**) ϵ=1.6, and (**l**) ϵ=1.8 for the 15 February 2020–15 August 2020 time window; and (fourth row) (**m**) ϵ=1.2, (**n**) ϵ=1.4, (**o**) ϵ=1.6, and (**p**) ϵ=1.8 for the 22 June 2020–22 December 2020 time window.

**Table 1 entropy-25-00846-t001:** The neck (nodes colored in red) between clusters in simple networks, and how they can be identified from the Fiedler vector, which is the eigenvector ${\vec{u}}_{1}$ associated with the first non-zero eigenvalue $\lambda_{1}$ of the graph Laplacian L.

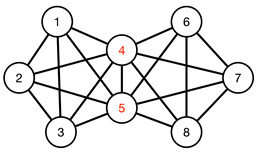	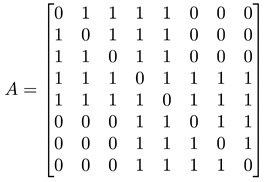	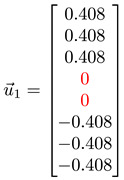
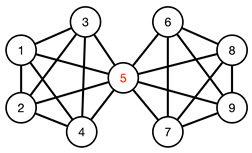	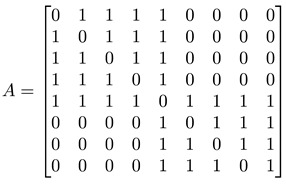	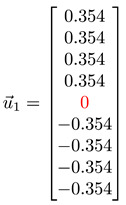
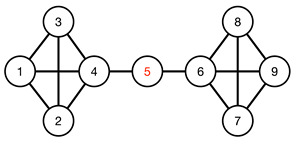	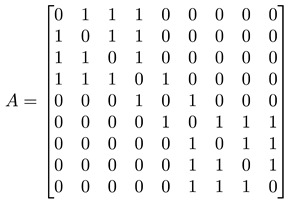	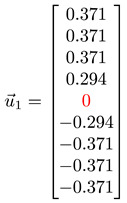
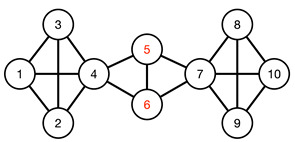	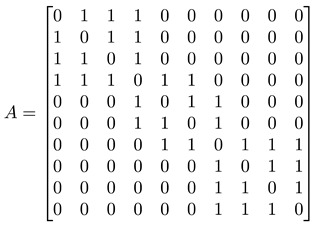	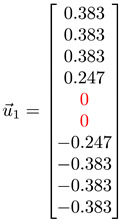
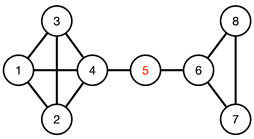	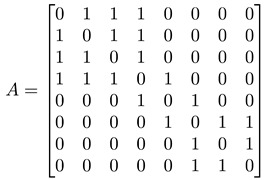	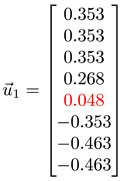

**Table 2 entropy-25-00846-t002:** The start and end dates of the four periods used to study the September 2018 mini-crash and March 2020 crash on the TWSE, the March 2020 crash on the SGX, and the February 2020 crash of the S&P 500.

Crash	Start Date	End Date
Sep 2018 TWSE mini-crash	1 April 2018	30 April 2019
Mar 2020 TWSE crash	1 August 2019	30 September 2020
Mar 2020 SGX crash	1 August 2019	30 April 2021
Mar 2020 S&P 500 crash	1 June 2019	31 December 2020

## Data Availability

All Python and Matlab scripts are available at https://doi.org/10.21979/N9/UCEELS, along with instructions on how to use them. This include reading the processed market data and perform the necessary computations to give the final results.

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
