# Peer review of "Laplacian Spectra of Persistent Structures in Taiwan, Singapore, and US Stock Markets"

_entropy, 2023, doi:10.3390/e25060846_

Round 1
Reviewer 1 Report
This article uses the spectral properties of the laplacian matrix of a graph to investigate the effect of three market crashes on the network's topology.
The eigenvalue gaps and the component of the Fiedler vector of the laplacian have been used by physicists, mathematicians, and computer scientists for many years.
The formalism is not new (even the similarities with degenerate quantum systems can already be found in the literature), and the results are marginal.
This paper is lengthy and suits a chapter in an educational book rather than a conventional publication.
Lacking novelty, incisive results, and brevity, I do not recommend it for
publication.
https://doi.org/10.1016/j.physa.2004.12.050
https://doi.org/10.1063/1.1985394
Author Response
Dear reviewer,
We have read your comments and made replies. Kindly find the attached file for reference.
with best regards,
Peter

Reviewer 2 Report
The present article is not a breakthrough paper, but it is clearly written, the methods are explored appropriately and the results corroborate the conclusions.
It would certainly have more impact if the authors had analysed a different stock exchange other than the Taiwanese which is not a top-tier stock exchange regarding market cap. Actually, it would be interesting to look at different markets as capitalisation is a proxy for liquidity which affects the collective behaviour of stocks.
The work finds relation to previous work on the collective dynamics of volume and the detection of collective and cross-sectional regime changes that took place during the sub-prime crisis. That shouuld be discussed.Graczyk MB, Duarte Queirós SM (2017) Intraday seasonalities and nonstationarity of trading volume in financial markets: Collective features. PLoS ONE 12(7): e0179198. https://doi.org/10.1371/journal.pone.0179198
and related references.
Author Response
Dear reviewer,
We have read your comments, and made necessary replies, and revisions. Kindly find the attached file for reference.
with warm regards,
Peter

Reviewer 3 Report
The work is interesting and presents important contributions to the discussion of the theme. However, there are some opportunities for improvement, which I will list below:
1. The abstract and conclusion should incorporate the limitations found; the practical implications; and indications for future research;
2. The formulas need to be centralized and numbered;
3. I suggest reducing the number of figures;
4. The explanatory notes for the figures are long. Revise all of them, and incorporate the explanations into the text.
5. Include at the end a list of acronyms used in the text.
6. Review the address of the sites indicated in the Supplementary Materials section.
7. 7. Revise the references cited with the format indicated in the instructions for authors.
Good review.
Author Response

(The authors gave the same response as above.)

Round 2
Reviewer 1 Report
I appreciate the Authors' effort to improve the quality of their manuscript, but I'm still not convinced of the originality and significance of their message. Applying spectral methods to infer topological transitions in graphs is not new. One defines a convenient matrix, e.g., the Laplacian, feeds in data and comments on the results.
A different story would be if we could get some hints on the precursors of topological transitions, as Thurner and colleagues do here: https://doi.org/10.1038/s41598-020-57751-y
As for the quantum analogy, the newly cited ref [38] uses it, although it is never mentioned explicitly. There was no need to cite another field in that context.
All in all, I still think the manuscript is not something I would personally publish. This does not mean that an editorial decision in favor may be excluded.
Reviewer 3 Report
Dear authors
In this second round of revision, I could see that the authors have done extensive work of revision and incorporating the suggestions pointed out by the reviewers. In the current form, there is no point of improvement. I believe that it has the minimum conditions for publication.
Success in new researches
Reviewer.